# A renal YY1-KIM1-DR5 axis regulates the progression of acute kidney injury

Chen Yang[1,6], Huidie Xu[1,6], Dong Yang[1,2], Yunhao Xie[2], Mingrui Xiong[1], Yu Fan[2], XiKai Liu[2], Yu Zhang[1], Yushuo Xiao[1], Yuchen Chen[1], Yihao Zhou[2], Liangliang Song[1], Chen Wang[1], Anlin Peng[3], Robert B. Petersen[4], Hong Chen[1] ✉, Kun Huang[1,5] ✉ & Ling Zheng[2] ✉

Acute kidney injury (AKI) exhibits high morbidity and mortality. Kidney injury molecule-1 (KIM1) is dramatically upregulated in renal tubules upon injury, and acts as a biomarker for various renal diseases. However, the exact role and underlying mechanism of KIM1 in the progression of AKI remain elusive. Herein, we report that renal tubular specific knockout of *Kim1* attenuates cisplatin- or ischemia/reperfusion-induced AKI in male mice. Mechanistically, transcription factor Yin Yang 1 (YY1), which is downregulated upon AKI, binds to the promoter of *KIM1* and represses its expression. Injury-induced KIM1 binds to the ECD domain of death receptor 5 (DR5), which activates DR5 and the following caspase cascade by promoting its multimerization, thus induces renal cell apoptosis and exacerbates AKI. Blocking the KIM1-DR5 interaction with rationally designed peptides exhibit reno-protective effects against AKI. Here, we reveal a YY1-KIM1-DR5 axis in the progression of AKI, which warrants future exploration as therapeutic targets.

Acute kidney injury (AKI), characterized by a sharp decrease in the glomerular filtration rate (GFR), causes high morbidity and mortality. Annually, approximately 13.3 million patients are newly diagnosed with AKI, leading to 1.7 million deaths[1]. Moreover, patients that have incomplete recovery from AKI are prone to subsequent development of chronic kidney disease (CKD) and eventually irreversible end-stage renal disease (ESRD)[2].

Kidney injury molecule-1 (KIM1, also known as HAVCR1/TIM1) is a transmembrane glycoprotein that belongs to the T-cell immunoglobulin and mucin domain family[3,4]. Structurally, KIM1 consists of an immunoglobulin variable Ig-like (Ig V) domain, mucin domain, transmembrane domain, and cytosolic domain[4]. KIM1 is primarily expressed in kidney and drastically upregulated in injured renal tubules[5]. Therefore, KIM1 is regarded as a sensitive and specific biomarker for

kidney injury[3,4], which has been approved by the FDA to evaluate nephrotoxicity[6]. However, there is limited information about the underlying mechanism that dramatically up-regulates KIM1 upon injury, and clarifying its role in injury will help elucidate AKI pathogenesis.

Apoptosis of renal tubules is a pathological characteristic of kidney injury[7]. Animal models and clinical data suggest the critical role of tubular apoptosis in AKI progression[8,9]. Furthermore, several reno-protective agents ameliorate AKI by diminishing tubular apoptosis[10,11]. Simultaneously tubule-specific knockout of two apoptotic genes, *Bax* and *Bak*, attenuate tubular apoptosis and ameliorate AKI[12]. Cellular apoptosis initiated by the multimerization of death receptor 5 (DR5) results in the recruitment of Fas-associated death domain (FADD) to induce formation of the death-inducing signaling complex (DISC),

[1]School of Pharmacy, Tongji Medical College and State Key Laboratory for Diagnosis and Treatment of Severe Zoonotic Infectious Diseases, Huazhong University of Science and Technology, Wuhan 430030, China. [2]Hubei Key Laboratory of Cell Homeostasis, Frontier Science Center for Immunology and Metabolism, College of Life Sciences, Wuhan University, Wuhan 430072, China. [3]Department of Pharmacy, The Third Hospital of Wuhan, Tongren Hospital of Wuhan University, Wuhan 430070, China. [4]Foundational Sciences, Central Michigan University College of Medicine, Mt. Pleasant, MI 48859, USA. [5]Tongji-RongCheng Biomedical Center, Tongji Medical College, Huazhong University of Science and Technology, Wuhan 430030, China. [6]These authors contributed equally: Chen Yang, Huidie Xu. ✉e-mail: hongchen2017@hust.edu.cn; kunhuang@hust.edu.cn; lzheng@whu.edu.cn

leading to activation of the caspase cascade[13–15]. Whether KIM1 regulates apoptosis, especially DR5-mediated apoptosis in AKI, remains unknown.

Here, we show that renal tubular specific knockout of *Kim1* relieves the progression of both cisplatin and ischemia-reperfusion injury (IRI) induced AKI. KIM1 is negatively regulated by transcription factor Yin Yang 1 (YY1); upon injury, upregulated KIM1 binds and activates DR5 by promoting multimerization, thus aggravates the apoptosis of renal cells and exacerbates AKI. Rationally designed peptides that block the interaction between KIM1 and DR5 protect against AKI.

## Results

### KIM1 is dramatically upregulated and senses AKI

KIM1 is a type I transmembrane protein that is mainly expressed in renal tubular epithelial cells (Fig. 1a). In cisplatin- and unilateral IRI (uIRI)-induced AKI mouse models, *Kim1* was the most sensitive biomarker of injury in comparison to *Cyr61, Nhe3, Fabp4, Il18* and *Ngal* (Fig. 1b & Supplementary Fig. 1a). The protein level of KIM1, mainly in injured renal tubules, was significantly upregulated at Day 3 in both cisplatin- and uIRI-induced injury (Fig. 1c, d & Supplementary Fig. 1b, c). After treating HK-2 cells with different doses of cisplatin for 24 h, the transcriptional and protein levels of KIM1 were upregulated (Supplementary Fig. 1d, e). Based on the results of this experiment, 5 μg/mL cisplatin was used in the rest of the in vitro experiments. To investigate whether other stimuli, such as cytokine production, NF-κB activation or oxidative stress, alone or in combination with cisplatin, affect *KIM1* expression, HK-2 cells were treated with IL-6, or TNF-α, or $H_2O_2$ with or without cisplatin. The mRNA level of *KIM1* was upregulated in response to cisplatin, or TNF-α or IL-6 or $H_2O_2$ per se (Supplementary Fig. 1f). Moreover, IL-6 exhibited synergetic effect with cisplatin on *KIM1* upregulation (Supplementary Fig. 1f).

### KIM1 aggravates inflammation and apoptosis in renal tubular epithelial cells

To investigate the role of KIM1 in inflammation and apoptosis, KIM1 overexpression or knockout HK-2 cells were generated (Supplementary Fig. 1g–j). Overexpression of KIM1 aggravated the cytotoxicity of cisplatin (Fig. 1e), while knockout of KIM1 relieved cisplatin-induced cell death (Fig. 1f). Moreover, mRNA levels of the inflammatory factors *IL6, TNFA, CXCL2*, and *CXCL10*, as well as cleaved PARP-1 (c-PARP-1) and phosphorylated p53 (p-p53) levels were also upregulated in KIM1 overexpressing cells and downregulated in KIM1 knockout cells following cisplatin injury (Fig. 1g–j). An increased percentage of apoptotic cells were detected when KIM1 was overexpressed (Cis *vs* KIM1 + Cis, 36.53 ± 2.46% *vs* 65.20 ± 0.10%), while a decreased percentage of apoptotic cells was observed after knockout of KIM1 (Cis *vs* KIM1 KO + Cis, 57.07 ± 7.22% *vs* 13.80 ± 0.54%) following cisplatin injury (Fig. 1k, l). Meanwhile, the number of TUNEL+ cells was significantly increased in KIM1 overexpressing cells and was reduced in KIM1 knockout cells after cisplatin injury (Fig. 1m–p).

### Transcription factor YY1 regulates *KIM1* expression

There is limited information about the factors that contribute to KIM1 upregulation following injury. To determine the transcription factors (TFs) that might regulate *KIM1* expression, prediction databases, including JASPAR, hTFtarget, and human TFDB 3.0 were used (Fig. 2a). 23 intersecting TFs that potentially regulate *KIM1* expression were selected and their abundance in kidney was further assessed using the Human Protein Atlas (https://www.proteinatlas.org/) (Supplementary Fig. 2a), and their expression levels were examined in the kidneys of cisplatin-injured mice (Fig. 2b). *STAT3, STAT1*, and *YY1*, which are abundant in the kidney (nTPM > 20) and were significantly altered after cisplatin injury, were further evaluated. Luciferase reporter assays indicated that STAT3 showed mild promotion, STAT1 exhibited no

obvious effect, while YY1 showed significant inhibition on *KIM1* transcription (Fig. 2c).

In cisplatin- and uIRI-injured mice, a time-course study suggested that *Yy1* was dramatically decreased at Day 1 after injury when significantly increased *Kim1* was found. *YY1* started to restore its transcriptional level at Day 3 after the injury when less *Kim1* was found in uIRI-injured mice (Supplementary Fig. 2b). Together, the two AKI models showed a similar expression pattern for *Kim1* and *Yy1* following injury. A significantly higher *Kim1* level and lower *Yy1* level were observed at Days 1 and 3 after uIRI injury in comparison to cisplatin injury (Supplementary Fig. 2b). Nevertheless, a negative correlation between the mRNA levels of *Yy1* and *Kim1* was found at Day 3 after the injury in both models (Supplementary Fig. 2c, d). The YY1 protein level was consistently downregulated in the kidneys of cisplatin- and uIRI-injured mice at Day 3 after the injury (Fig. 2d–g).

Moreover, in a time-course study on cisplatin-treated HK-2 cells, the *YY1* level was significantly reduced at 3 h after treatment and remained at a low level for 12 h, starting to recover at 24 h after the injury. In contrast, the *KIM1* level was upregulated at 6 h, peaking at 12 h, and beginning to decrease at 24 h after the injury (Fig. 2h), which suggests that the expression of *YY1* is ahead of that of *KIM1* upon injury.

YY1 has been implicated in DNA damage recognition in β-cells[16]. To investigate whether YY1 responds to DNA damage in renal cells, HK-2 cells were treated with etoposide, a classic inducer of DNA damage[17]. The DNA damage marker *P21* was upregulated at 5 min, while *YY1* was decreased at 10 min after etoposide treatment (Supplementary Fig. 2e).

We further evaluated how YY1 regulated KIM1 expression. Overexpressing YY1, or activating YY1 with its agonist eudesmin[18], suppressed KIM1 expression (Fig. 2i–l), whereas knockdown of YY1 increased KIM1 expression at both the mRNA and protein level (Fig. 2m, n). To investigate whether YY1 binds to the promoter of *KIM1*, we performed ChIP assays in HK-2 and mouse primary tubular epithelial cells (mPTECs) with or without cisplatin and found that YY1 bound to the *KIM1* promoter at the P3 and P4 regions (Fig. 2o, p). Enriched YY1 was consistently found at multiple regions of the *KIM1* promoter in lymphoblastoid cells, as suggested by ChIP-sequencing data from the Gene Transcription Regulation Database (http://gtrd.biouml.org/) (Supplementary Table 1)[19].

### YY1 protects against AKI in vitro and in vivo

Because YY1 negatively regulated KIM1 following renal injury, we further investigated whether YY1 protected against AKI. Overexpression of YY1 or eudesmin pre-treatment protected HK-2 cells against cisplatin-induced injury and suppressed the mRNA levels of the inflammatory factors *IL6* and *CXCL2* (Fig. 3a–c), while knockdown of YY1 exacerbated cisplatin-induced cell death and the inflammatory response (Fig. 3d, e). In cisplatin-induced AKI mice, eudesmin treatment consistently promoted the expression of YY1 and inhibited KIM1 expression (Fig. 3f, g), reduced serum creatinine and urea nitrogen (BUN) levels and attenuated pathological injury (Fig. 3h–j).

### KIM1 binds DR5 and activates its downstream caspase cascade

To uncover KIM1 downstream effectors, an affinity-based mass-spectrometry was performed to identify KIM1 binding partners in HEK293T cells (Supplementary Fig. 3a). The TNF receptor superfamily member 10b (TNFRSF10B; also known as DR5) was among the top 20 scored candidate binding partners (Supplementary Table 2). We further confirmed the binding between KIM1 and DR5 using Co-IP and fluorescence resonance energy transfer (FRET) assays. A Co-IP study demonstrated interaction between KIM1 and DR5, with or without DR5 overexpression, at both physiological conditions and under cisplatin injury (Fig. 4a, b). The KIM1-DR5 interaction was supported by FRET assays, and the interaction was enhanced following cisplatin treatment, as indicated by robustly upregulated

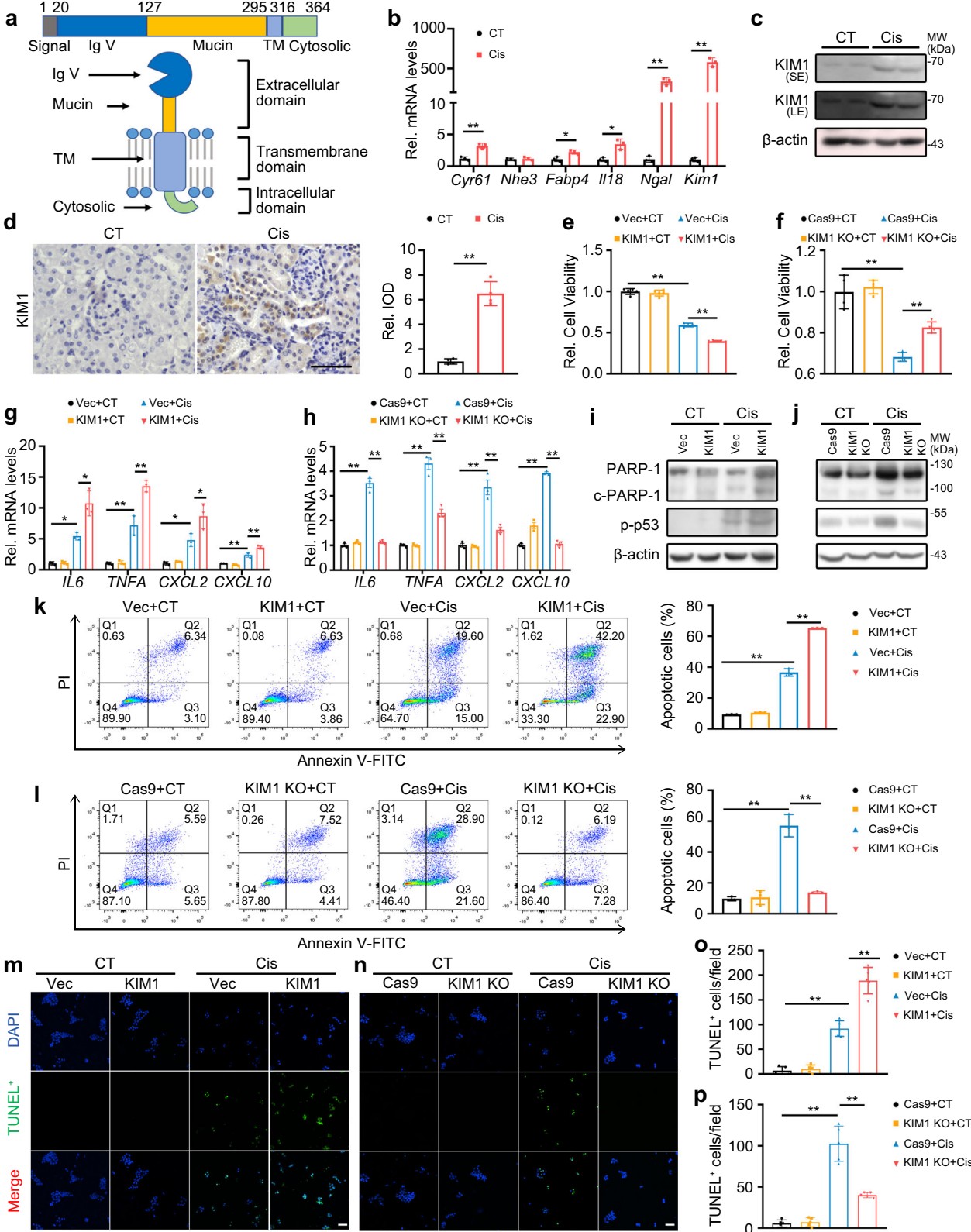

FRET intensity (Fig. 4c–e). Moreover, gradient overexpression of KIM1 demonstrated gradually enhanced binding between KIM1 and DR5 (Supplementary Fig. 3b). While FRET assays confirmed the binding between KIM1-CFP and DR5-YFP, it was difficult to probe endogenous KIM1-DR5 interaction by FRET, because the large sizes of primary and secondary antibodies are beyond the detection limit of FRET[20,21]. As future studies, endogenous FRET assays may be performed by engineering cells using CRISPR/Cas9 mediated knock-

in system[22], so that KIM1 and DR5 are labeled endogenously with CFP or YFP, respectively.

Interestingly, DR5 was upregulated while its ligand TRAIL was unchanged in the two AKI mouse models (Fig. 4f & Supplementary Fig. 3c–f). Since DR5 plays crucial roles in mediating apoptosis, we suspected that the KIM1-DR5 interaction contributes to the apoptotic-promoting effects of KIM1 in AKI. DR5 activation leads to clustering of death domains and the subsequent DISC formation[14], which is

**Fig. 1 | KIM1 is dramatically upregulated in a cisplatin-induced AKI model, aggravating inflammation and apoptosis in renal tubular epithelial cells.**
**a** Structural scheme of KIM1 domains. Signal signal peptide, Mucin mucin-containing domain, TM transmembrane domain, CytD cytosolic domain. **b** mRNA levels of several AKI biomarkers in mouse kidneys at Day 3 after cisplatin (Cis) injury. $n = 3$ mice per group. **c** KIM1 protein level in mouse kidneys at Day 3 after cisplatin injury. LE/SE, long/short exposure. $n = 2$ mice per group, each experiment was repeated at least three times independently with similar results obtained. **d** Representative images of immunohistochemistry staining and quantitation of KIM1 in renal sections at Day 3 after cisplatin injury. Scale bar, 50 μm. $n = 4$ mice per group. **e, f** MTT assays to assess the effects of KIM1 overexpression (**e**) or knockout (**f**) in HK-2 cells with or without 24 h cisplatin stress. Vec, pRK-5'Flag; KIM1, pRK-5'Flag-KIM1; Cas9, lenti-CRISPR/Cas9; KIM1 KO, lenti-CRISPR/Cas9-based KIM1 knockout. CT, without cisplatin treatment. $n = 4$-5 biological samples per group, each experiment was repeated at least three times independently with similar results obtained. **g, h** qPCR of inflammatory factors with KIM1 overexpression (**g**) or knockout (**h**) in HK-2 cells with or without 24 h cisplatin stress. $n = 3$ biological samples per group, each experiment was repeated at least three times independently with similar results obtained. **i, j** Western blots of apoptotic molecules in KIM1 overexpression (**i**) or knockout (**j**) HK-2 cells with or without 24 h cisplatin stress. Each experiment was repeated at least three times independently with similar results obtained. **k, l** Representative flow cytometry results and quantitative data of KIM1 overexpression (**k**) or knockout (**l**) HK-2 cells with or without 24 h cisplatin stress analyzed by Annexin V-FITC and propidium iodide (PI) labeling. Quantitative data provided the average and standard deviation from three independent experiments (percentage of apoptotic cells was calculated by Annexin-V positive cells (Q2 + Q3)). Each experiment was repeated at least three times with representative results shown. **m–p** Representative images of TUNEL assay (**m, n**) and quantitative results (**o, p**) in KIM1 overexpression (**m, o**) or knockout (**n, p**) groups. $n = 5$ biological samples per group, each experiment was repeated at least three times independently with similar results obtained. Scale bar, 50 μm. Data shown as mean ± SD. Two-tailed unpaired Student's $t$-test was used for two experimental groups and one-way ANOVA for multiple experimental groups without adjustment. $^*P < 0.05$; $^{**}P < 0.01$. Exact $P$ values are provided in Source Data.

composed of FADD and caspases cascade[13]. Overexpression of KIM1 activated the caspase cascade downstream of DR5 as indicated by upregulated cleavage of caspases 3, 8 and 9 (Fig. 4g), while knockdown of KIM1 inhibited this activation (Fig. 4h). Moreover, in cisplatin-injured HK-2 cells and mouse kidneys, partial co-location of KIM1 and DR5 was observed (Fig. 4i, j). DR5 knockdown abolished KIM1 overexpression induced cell death and apoptosis (Fig. 4k–m). These results indicated that KIM1 aggravated kidney injury in a DR5-dependent manner, in which the interaction between KIM1 and DR5 plays an important role.

## KIM1 promotes DR5 multimerization via binding to its ECD domain

DR5 consists of an extracellular domain (ECD), a transmembrane domain (TMH), and an intracellular domain (CytD) (Fig. 5a)[14]. The apoptosis-promoting effects of DR5 largely depend on its multimerization[14,23], upon which, clustered cytoplasmic domains of DR5 interact with the adapter FADD, subsequently recruiting effectors like caspase 8 to form DISC (Fig. 5a)[14]. Since KIM1 activated the DR5 downstream signaling pathway (Fig. 4), we investigated whether KIM1 affected DR5 multimerization.

First, we constructed a FRET system using CFP- and YFP-tagged DR5. Fluorescence detection suggested that KIM1 promoted the multimerization of DR5 after cisplatin injury, as indicated by a robust upregulation of FRET intensity when KIM1 was overexpressed; while a reduction in FRET intensity was observed when KIM1 was knocked out (Fig. 5b–e & Supplementary Fig. 3g). Consistently, in acceptor photobleaching assays, receptor (DR5-YFP) photobleaching led to enhanced donor (DR5-CFP) intensity (Supplementary Fig. 3h). Moreover, photobleaching of DR5-YFP caused a much higher increase in DR5-CFP intensity upon cisplatin injury and KIM1 overexpression. When KIM1 was knocked out, photobleaching of DR5-YFP gave a lower DR5-CFP intensity even after cisplatin injury (Fig. 5f, g). Similar conclusions were drawn from FRET efficiency calculations (Fig. 5h, i). Next, we performed native PAGE electrophoresis, the results indicated that after cisplatin injury, KIM1 overexpression promoted the formation of higher-order DR5 oligomers, while knockout of KIM1 suppressed oligomer formation (Fig. 5j, k).

We further mapped the sites of interaction between KIM1 and DR5. Alphafold2 simulation suggested that ECD deletion abolished KIM1-DR5 interaction as indicated by decreased $\Delta^i G$ and interface area, while deletion of TMH or CytD showed much less impact (Supplementary Table 3). Further domain mapping experiments gave similar conclusions, in which KIM1 bound with full length DR5, DR5 ΔTMH, and DR5 ΔCytD, but not with DR5 ΔECD (Fig. 5l). On the other hand, the Ig V domain, but not Mucin or CytD domain of KIM1, bound with DR5 (Fig. 5m).

To test whether blocking DR5 multimerization protect against renal cell injury, nystatin was used, which has been reported to disrupt lipid rafts mediated DR5 multimerization[24]. MTT assays suggested that nystatin protected against cisplatin injury in HK-2 cells with or without KIM1 overexpression (Supplementary Fig. 4a–c). Further FRET assays also demonstrated that nystatin inhibited DR5 multimerization after cisplatin injury in HK-2 cells, with or without KIM1 overexpression (Supplementary Fig. 4d). Moreover, nystatin reduced the formation of higher-order DR5 oligomers in HK-2 cells as demonstrated by native PAGE electrophoresis (Supplementary Fig. 4e). Interestingly, no obvious effect on DR5 multimerization was found in atorvastatin or perifosine treated HK-2 cells (Supplementary Fig. 4f, g), which have been reported to respectively either inhibit or promote lipid raft accumulation[25–27], suggesting that nystatin may affect DR5 multimerization independent of its effects on lipid rafts.

## Renal tubular specific knockout of *Kim1* relieves cisplatin-induced AKI

KIM1 is mainly expressed in tubules (Fig. 1d & Supplementary Fig. 1c), so we generated renal tubular specific *Kim1* knockout (*Kim1*[Ksp] KO) mouse by mating *Kim1* floxed mouse with Ksp-Cre transgenic mouse (Fig. 6a & Supplementary Fig. 5a). KIM1 was significantly reduced in the kidney of *Kim1*[Ksp] KO mice compared to wildtype (WT) mice (Supplementary Fig. 5b). Co-staining KIM1 with lotus tetragonolobus lectin (LTL) and peanut agglutinin (PNA) demonstrated successful knockout of KIM1 in the proximal and distal tubules (Supplementary Fig. 5c, d).

Following cisplatin injury, compared with the WT mouse, the *Kim1*[Ksp] KO mouse at Day 3 after the injury exhibited decreased mRNA and protein levels of KIM1, as well as decreased serum creatinine and BUN levels (Fig. 6b–e). The *Kim1*[Ksp] KO mouse also showed significantly less severe tubular injury at Day 3 after the injury (Fig. 6f). Moreover, there was also decreased mRNA level of another kidney injury biomarker, *Ngal*, in the injured *Kim1*[Ksp] KO mouse (Fig. 6g). Consistently, the injured *Kim1*[Ksp] KO mouse showed decreased transcriptional levels of inflammatory factors *Tnfa*, *Mcp1*, *Il6* and *Cxcl10*, and apoptotic molecule *Bax*, as well as upregulated expression of the anti-apoptotic *Bcl2*(Fig. 6h, i). Moreover, mRNA levels of fibrotic factors *Tgfb1*, *Col1a1* and *Fn1* were also decreased in injured *Kim1*[Ksp] KO mouse (Fig. 6j). Meanwhile, injured *Kim1*[Ksp] KO mice showed decreased serum IL-6 level and fewer infiltrating macrophages (F4/80$^+$) (Supplementary Fig. 6a, b). In addition, we found that in *Kim1*[Ksp] KO mice, there was inhibition of the injury-induced caspase cascade activation, as indicated by significantly decreased cleaved caspases 3, 8, and 9 (Fig. 6k). Consistently, TUNEL assay indicated reduced cell death in injured *Kim1*[Ksp] KO mice (Fig. 6l). Crucially, there was decreased formation of higher-order DR5 oligomers suggesting a reduction in multimerization in injured *Kim1*[Ksp] KO mice (Fig. 6m). Together, these data suggest

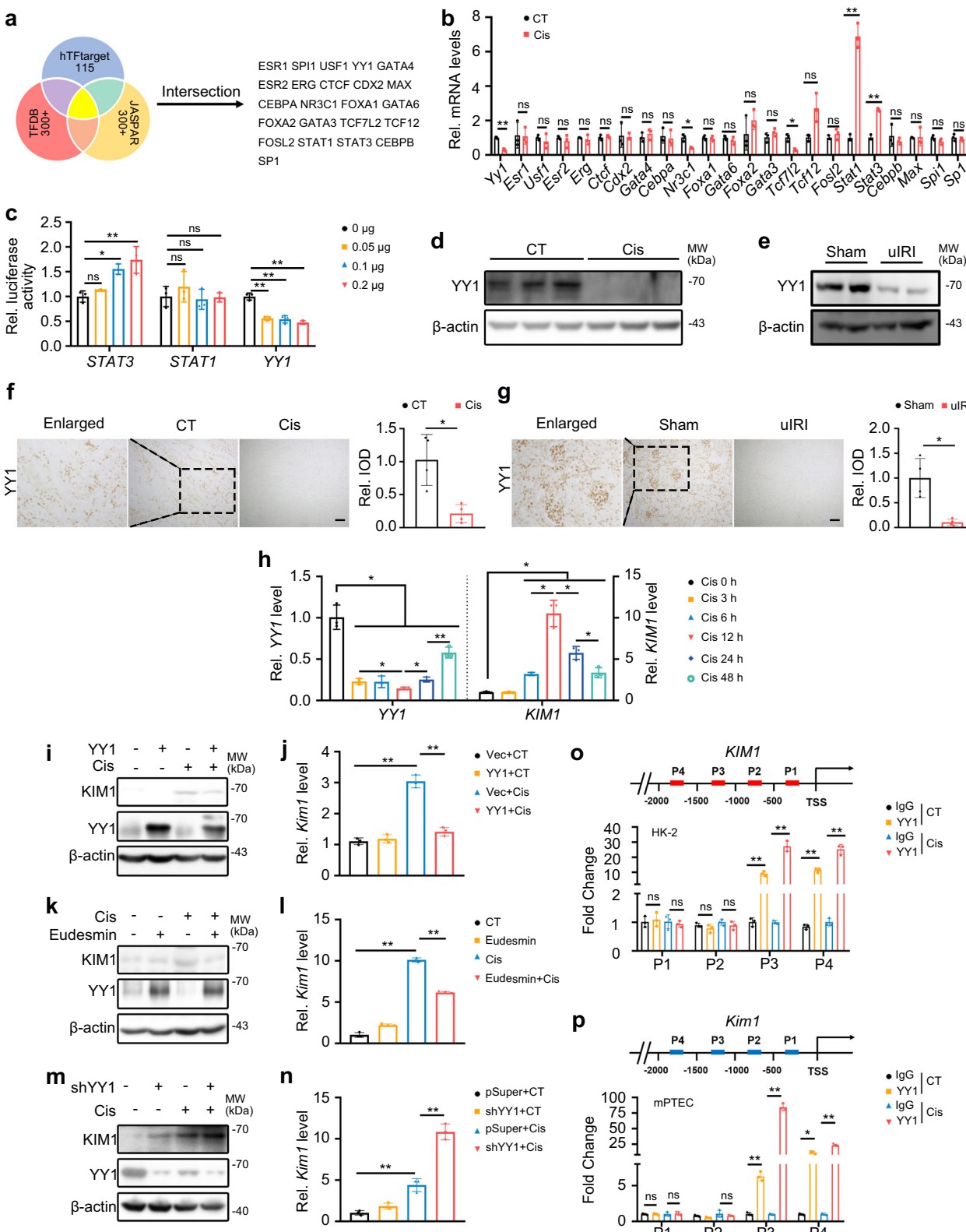

that renal tubular specific knockout of *Kim1* attenuated cisplatin-induced AKI.

**Renal tubular specific knockout *Kim1* ameliorates renal IR injury**

A bilateral IRI (bIRI) model was also used to study the role of KIM1 in AKI. Reduced mRNA and protein levels of KIM1 were observed in injured *Kim1^Ksp* KO mice compared to WT mice (Fig. 7a, b). Decreased serum creatinine and BUN levels, with lower tubular injury scores were found in *Kim1^Ksp* KO mouse at Day 1 after bIRI (Fig. 7c, e). As in cisplatin-induced AKI mice, the mRNA levels of the AKI biomarker *Ngal*, inflammatory factors *Tnfa*, *Mcp1*, *Il6*, *Cxcl2* and *Cxcl10*, apoptotic molecules *Fas* and *Bax*, and fibrotic factors *Col1a1* and *Fn1* were decreased in bIRI-injured *Kim1^Ksp* KO mice (Fig. 7f–i). Decreased serum IL-6 and fewer infiltrating macrophages (F4/80⁺) were found in bIRI-injured *Kim1^Ksp* KO mice (Supplementary Fig. 6c, d). Moreover, tubular KIM1 knockout inhibited the injury-induced activation of caspase

**Fig. 2 | YY1 is downregulated in AKI models and negatively regulates *KIM1* expression. a** Database screening for potential *KIM1*-regulating transcriptional factors (TFs). 23 TFs were found from intersections of JASPAR, Human TFDB and hTFtarget. **b** qPCR of 23 potential *KIM1*-regulating TFs in mouse kidneys at Day 3 after cisplatin (Cis) injury. $n = 3$ mice per group. **c** Luciferase reporter assays for the effects of STAT3, STAT1 and YY1 on *KIM1* promoters. STAT3, prK-5'Flag-STAT3; STAT1, prK-5'Flag-STAT1; YY1, prK-YY1-3'HA; $n = 3$ biological samples per group, each experiment was repeated at least three times independently with similar results obtained. **d, e** Protein levels of YY1 in mouse kidneys at Day 3 after cisplatin injury (**d**) or **e** unilateral renal ischemia-reperfusion injury (uIRI). **d** $n = 3$ mice per group, **e** Sham, non-injury control, uIRI, unilateral ischemia-reperfusion injury. $n = 2$ mice per group. Each experiment was repeated at least three times independently with similar results obtained. **f, g** Representative images of YY1 immunohistochemistry staining with quantitative analysis at Day 3 after cisplatin injury (**f**) or uIRI (**g**). Scale bar, 50 μm. Integrated option density, IOD. (**f, g**) $n = 4$ mice per group. **h** qPCR of *YY1* and *KIM1* in HK-2 cells treated with 5 μg/mL cisplatin for the indicated time. $n = 3$ biological samples per group, each experiment was repeated at least three times independently with similar results obtained. Two-tailed unpaired Student's *t*-test was used. **i, j** Protein (**i**) and mRNA (**j**) levels of KIM1 in YY1 overexpression groups with/without 24 h cisplatin. Vec, prK-3'HA; YY1, prK -YY1-3'

HA. **i** Each experiment was repeated at least three times independently with similar results obtained; **j** $n = 3$ biological samples per group, each experiment was repeated at least three times independently with similar results obtained. **k, l** Protein (**k**) and mRNA (**l**) levels of KIM1 in 1 μM eudesmin-treated groups with/without 24 h cisplatin. **k** Each experiment was repeated at least three times independently with similar results obtained; **l** $n = 3$ biological samples per group, each experiment was repeated at least three times independently with similar results obtained. **m, n** Protein (**m**) and mRNA (**n**) levels of KIM1 in *YY1* knockdown groups with/without 24 h cisplatin. Vec, prK-3'HA; YY1, prK-YY1-3'HA; pSuper, pSuper backbone; shYY1, pSuper-shYY1. CT, without cisplatin treatment. **m** Each experiment was repeated at least three times independently with similar results obtained; **n** $n = 3$ biological samples per group, each experiment was repeated at least three times independently with similar results obtained. **o, p** ChIP assays in HK-2 cells (**o**) and mPTECs (**p**) with/without 24 h cisplatin. **o, p** $n = 3$ biological samples per group, each experiment was repeated at least three times independently with similar results obtained. **p** Two-tailed unpaired Student's *t*-test was used. Data shown as mean ± SD. Two-tailed unpaired Student's *t*-test was used for two experimental groups, and one-way ANOVA for multiple experimental groups without adjustment. $^*P < 0.05$; $^{**}P < 0.01$; ns no significance. Exact *P* values are provided in Source Data.

cascade, as indicated by decreased levels of cleaved caspases 3, 8 and 9 (Fig. 7j). Consistently, TUNEL assay suggested a decrease in apoptosis in bIRI-injured *Kim1^Ksp* mice (Fig. 7k, l). Decreased formation of higher-order DR5 oligomers in renal tissues suggested that *Kim1* knockout attenuated DR5 multimerization (Fig. 7m).

### KIM1-DR5 interaction blockade peptides protect against AKI

Since KIM1 bound to DR5 activates the caspase cascade (Figs. 4 and 5), we hypothesized that interfering with the KIM1-DR5 interaction would protect against DR5-induced apoptosis and relieve AKI. To test this hypothesis, we performed an in silico design and screen for candidate blocking peptides.

High accuracy structure predictions by deep neural networks, like AlphaFold2 and RoseTTAFold, enable precise modeling of protein-protein interaction (PPI)[28], and provide fast in virtue screening. To screen and design peptides blocking the KIM1-DR5 interaction, we used Alphafold2 Colab (Fig. 8a). Among predicted binding sites, the 10 top-scored peptides derived from human KIM1/DR5 that potentially interfere with the KIM1-DR5 interaction were further screened (Supplementary Fig. 7a and Supplementary Table 4). Potential antagonistic effects against KIM1-DR5 interaction were simulated by inputting peptides, KIM1, and DR5. The $\Delta^i G$ (solvation free energy gain upon interface formation) of KIM1 and DR5 was calculated before and after antagonistic peptides input; and $\Delta\Delta^i G$ was calculated (Supplementary Table 4), in which three highest-scored antagonistic peptides (P1: KIM1 60-75 aa; P2: DR5 68-85 aa; P3: DR5 126-140 aa) exhibited high capacity for blocking the human KIM1-DR5 interaction (Supplementary Table 4). MTT assays demonstrated that peptide P1 protected against cisplatin injury in human HK-2 cells but was less effective in mouse TCMK-1 cells, whereas peptide P2 showed obvious protection in TCMK-1 and mPTEC cells (Fig. 8b and Supplementary Fig. 7b, c). Further molecular dynamic simulations demonstrated that peptide P2 showed a strong antagonistic effect against the mouse KIM1-DR5 interaction while peptide P1 preferentially blocked the human KIM1-DR5 interaction (Supplementary Tables 4 and 5). In cisplatin-injured TCMK-1 cells, peptide P2 inhibited transcription of *Il6*, *Fas* and *Bax*, promoted the transcription of anti-apoptotic *Bcl2*, and inhibited activation of the caspase cascade (Fig. 8c, d).

We further assessed the effects of peptide P2 in cisplatin-injured mice. By assessing renal function and pathology, we chose 1 μM of peptide P2 as the dosage for further animal studies (Supplementary Fig. 7d–f). To determine whether peptide P2 targets to cells overexpressing KIM1-DR5, we performed in vitro and in vivo studies. FITC-labeled P2 was added to HK-2 cells with or without cisplatin injury. The

FITC signal was found on the cell membrane of injured but not control cells, which was co-localized to injury-enhanced KIM1 and DR5 staining (Supplementary Fig. 8a). Dramatically increased fluorescence distribution in the kidney, but not in other organs, was found in cisplatin-injured mice injected with Cy7-labeled P2 (Supplementary Fig. 8b). Furthermore, P2 was labeled with 5'(6)-FAM, a green fluorescein with greater in vivo stability than FITC. Labeled P2 was co-localized with KIM1 and DR5 in the renal tubules of cisplatin-injured mice, but not in those of non-injured mice (Fig. 8e).

Next, we investigated the in vivo bioactivity of peptide P2 in cisplatin-injured mice (Supplementary Fig. 8c). Peptide P2 significantly ameliorated serum creatinine and urea nitrogen levels, rescued renal injury in cisplatin-injured mice (Fig. 8f–h). Moreover, peptide P2 significantly inhibited the activation of the caspase cascade, and attenuated cisplatin-induced apoptosis (Fig. 8i, j). Co-IP assays confirmed that peptide P2 blocked KIM1-DR5 interaction in cisplatin-injured kidney (Fig. 8k).

## Discussion

KIM1 is upregulated in tubules after AKI, and functions as a biomarker for a wide range of kidney diseases including AKI, DKD and renal cell carcinoma[29]. While most studies indicate that KIM1 plays a role in AKI, there is still controversy reports on its effects. On one hand, KIM1 is suggested to promote AKI progression. For example, KIM1 augments hypoxia-induced tubulointerstitial inflammation via uptake of small extracellular vesicles (sEV) derived from injured tubular epithelial cells (TECs); exogenously-applied hypoxic TEC-derived sEV localized to KIM1-positive tubules and aggravated tubulointerstitial inflammation in IR-injured mice[30]. On the other hand, KIM1 has also been suggested to protect against AKI in some circumstances. Deletion of the Mucin domain markedly impaired KIM1-mediated phagocytosis, resulting in increased proinflammatory cytokine production and decreased anti-inflammatory growth factor secretion; mice expressing this Mucin domain truncated KIM1 show severe impairment of renal function, inflammatory responses, and mortality in response to IRI- and cisplatin-induced AKI[31]. Our results suggest that KIM1 overexpression or upregulation aggravates AKI (Fig. 1). Moreover, KIM1 is mainly expressed in tubules[5,32], and by investigating *Kim1^Ksp* KO mice, we found that renal tubular specific *Kim1* knockout ameliorated cisplatin- or IRI-induced AKI (Figs. 6 and 7).

The rapid upregulation of KIM1 upon injury makes it a sensitive marker to renal injury, however, the factors that drive this fast upregulation remain unclear. A recent study established the enhancer and super-enhancer landscape by ChIP-seq in IR-injured kidneys, which

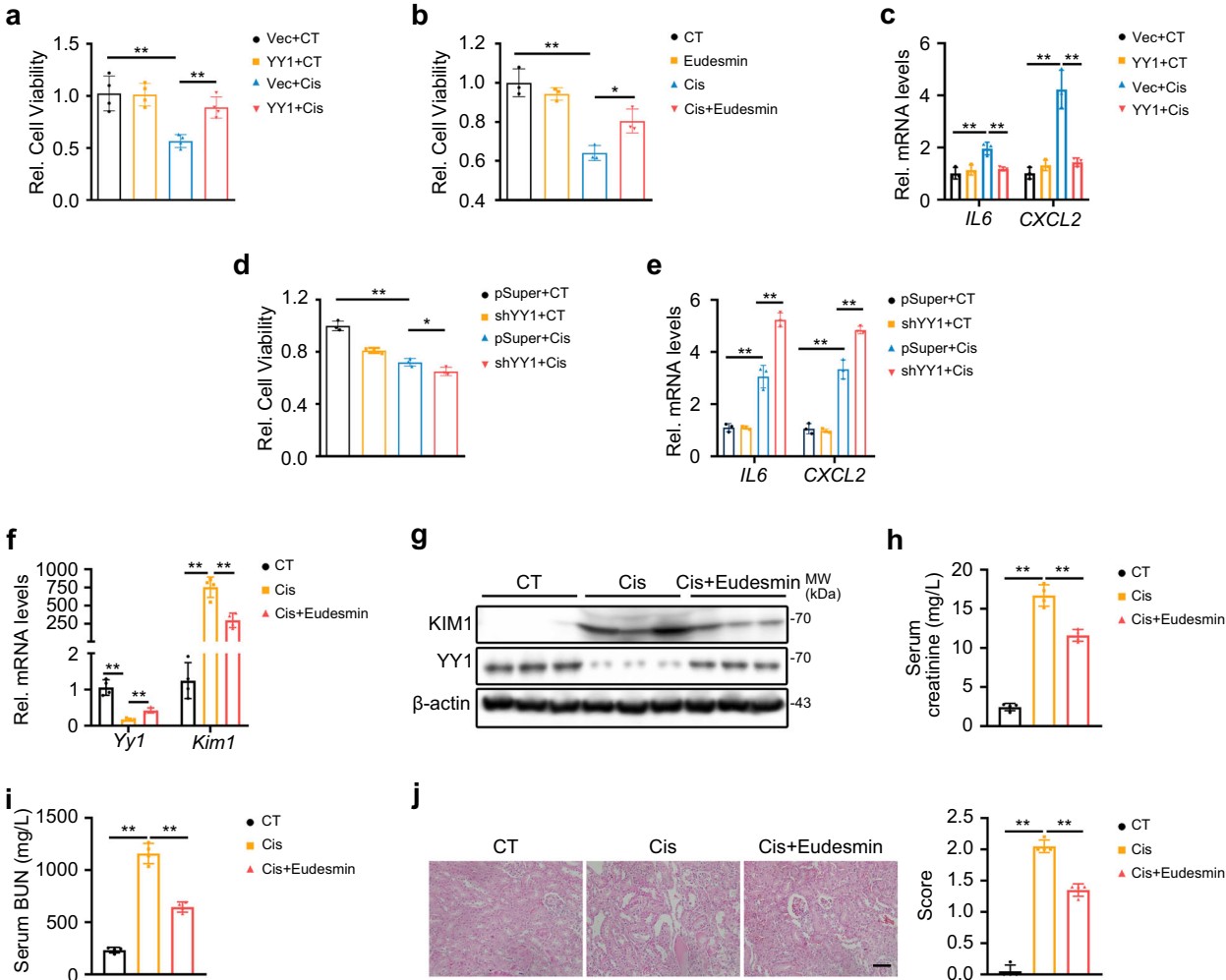

**Fig. 3 | YY1 protects against AKI in vitro and in vivo. a, b** MTT assays to assess the effects of YY1 overexpression (**a**) and eudesmin treatment (**b**) in HK-2 cells with/without 24 h cisplatin (Cis). Vec, pRK-3′HA; YY1, pRK-YY1-3′HA; CT, without cisplatin treatment. **a** $n = 4$ biological samples per group; **b** $n = 3$ biological samples per group, each experiment was repeated at least three times independently with similar results obtained. **c** qPCR analysis of *IL6* and *CXCL2* expression in YY1 overexpressing groups with/without 24 h cisplatin. $n = 3$ biological samples per group, each experiment was repeated at least three times independently with similar results obtained. **d, e** MTT assays (**d**) and qPCR analysis of *IL6* and *CXCL2* expression (**e**) in YY1 knockdown HK-2 cells with/without cisplatin injury. Vec, pRK-3′HA; YY1, pRK-YY1-3′HA; pSuper, pSuper backbone; shYY1, pSuper-shYY1; CT, without

cisplatin treatment. **d, e** $n = 3$ biological samples per group, each experiment was repeated at least three times independently with similar results obtained. **d** two-tailed unpaired Student's *t*-test was used. **f, g** qPCR (**f**) and Western blots (**g**) of KIM1 and YY1 in mouse kidneys at Day 3 after cisplatin injury with/without eudesmin treatment. **f** $n = 4$ mice per group; two-tailed unpaired Student's *t*-test was used. **g** $n = 3$ mice per group. **h–j** Serum creatinine level (**h**), serum urea nitrogen level (**i**) and pathological score (**j**) in mice at Day 3 after cisplatin injury with/without eudesmin treatment. Scale bar, 50 µm. **h–j** $n = 4$ mice per group; Data shown as mean ± SD. Two-tailed unpaired Student's *t*-test was used for two experimental groups, and one-way ANOVA for multiple experimental groups without adjustment. $^*P < 0.05$; $^{**}P < 0.01$. Exact *P* values are provided in Source Data.

suggests that HNF4A, GR, STAT3 and STAT5 may bind to the enhancer and super-enhancer to cause enhancer dynamics and transcriptional changes during kidney injury[33]. In another bioinformatic study, Chk1 and STAT3 were implicated as potential KIM1-regulating transcription factors, however, experimental evidence for this is lacking[34]. Here, by combining the screening results of online prediction servers, we identified transcription factors STAT1, STAT3 and YY1 as *KIM1*-regulating candidates. Luciferase reporter assays and ChIP assays indicated that STAT3 mildly promoted KIM1 expression, while YY1 significantly inhibited KIM1 transcription by binding to the promoter of *KIM1* (Fig. 2c, o, p). Interestingly, YY1 has been reported to have protective effects against DKD by transcriptionally inhibiting TGF-β expression[18]. Together, our results revealed a negative regulatory role of YY1 on KIM1 in AKI, moreover, previous studies and our data all suggested a complex network behinds KIM1 regulation.

Increased apoptosis has been widely reported upon renal injury, and various reagents that inhibit inflammation, oxidative stress,

mitochondrial dysfunction and consequently inhibits renal apoptosis present protective effects against renal injury[35–37]. This highlights that apoptosis represents a key event in the onset and development of the renal diseases, and equally underlines how a primum movens that could indicate a promising and innovative therapeutic target has not yet been identified. Here, we found that DR5, a well-known apoptotic molecule that plays a deleterious role in scald burns caused renal injury[38], binds KIM1. Our findings suggested that KIM1 significantly enhanced DR5 multimerization without affecting DR5 levels upon injury, both in vitro and in vivo (Figs. 5, 6m and 7m); even though DR5 was upregulated following cisplatin- and IRI-induced AKI (Fig. 4f). In silico and domain mapping studies suggested that the KIM1 Ig V domain binds with the ECD domain of DR5 (Fig. 5l, m & Supplementary Table 3). Knock-in mice that express KIM1/DR5 construct(s) with compromised interface between DR5 and KIM1 will be an interesting future direction. The classic DR5 ligand TRAIL has been reported to bind the ECD of DR5, and subsequently activate DR5-mediated

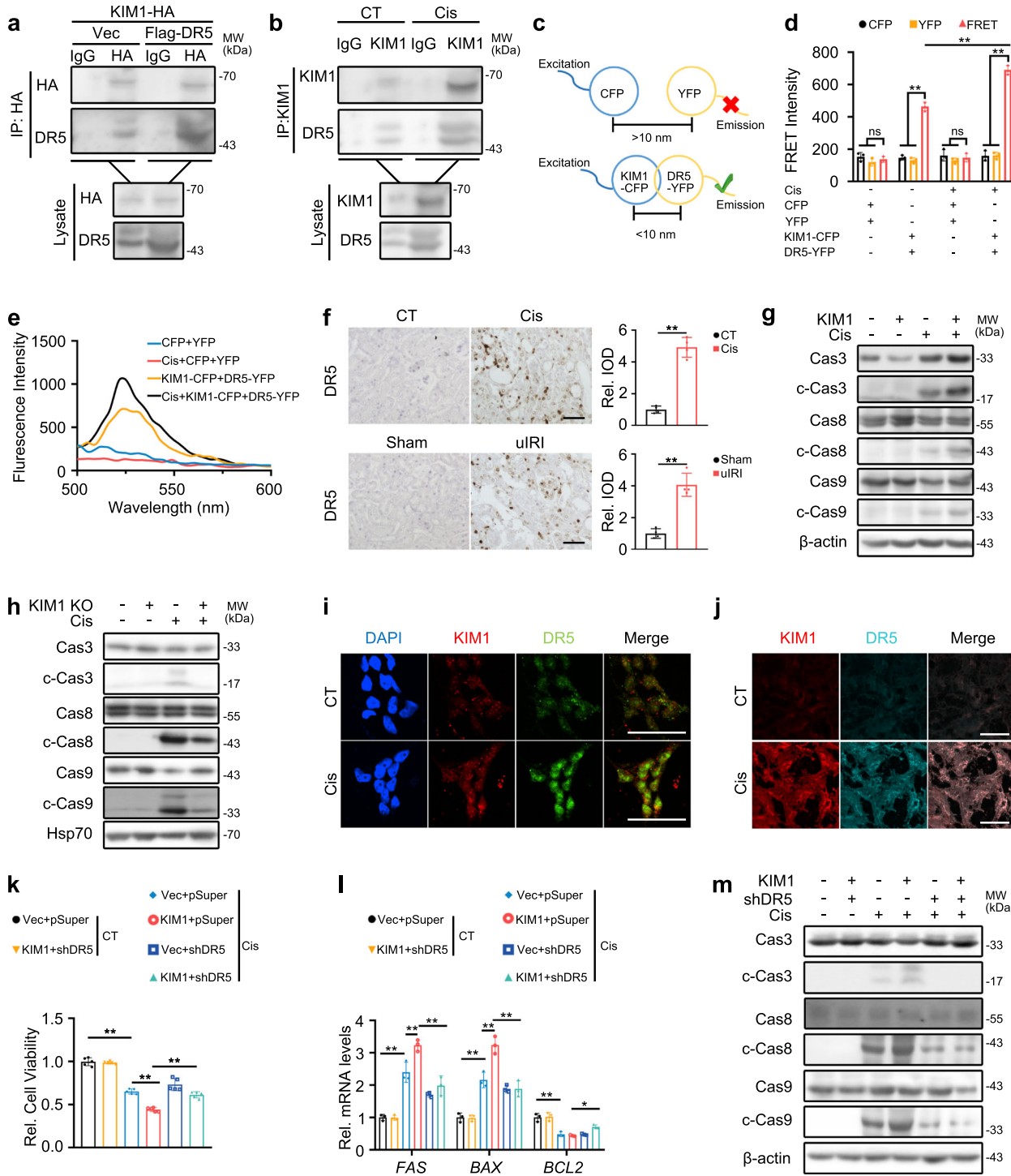

apoptosis by promoting DR5 multimerization[14]. However, no change in the level of TRAIL was observed after renal injury (Supplementary Fig. 3c–f). Therefore, we reasoned that injury-induced KIM1 may play a TRAIL-like role that promotes apoptosis in AKI progression. Considering that recombinant TRAIL has been proposed as a promising therapy for various cancers by arousing apoptosis[39], it will be interesting to study whether KIM1 has similar anti-cancer effects.

The YY1-KIM1-DR5 axis unveiled in this study suggested multiple potential intervention targets for AKI therapy. First, given the deleterious role of upregulated KIM1 in AKI, and the beneficial effects of upregulation/activation of YY1 on AKI (Fig. 3), our study supports further investigation into the development of YY1 agonists. Secondly, nystatin[23], a previously reported DR5 multimerization

disruptor whose exact working mechanism requires future study, inhibited the multimerization of DR5 and protected against cisplatin-induced cell death (Supplementary Fig. 4a–e). Therefore, developing specific and effective DR5 inhibitors could be further explored for AKI treatment. Lastly, our data suggested that the KIM1-DR5 interaction played a crucial role in exacerbating AKI (Figs. 4 and 5). Consequently, through in silico molecular simulations and rational design, we obtained a set of antagonistic peptides that block KIM1-DR5 interaction, among which peptide P2 significantly blocked KIM1-DR5 interaction and exhibited reno-protect effects in vitro and in vivo (Fig. 8 and Supplementary Fig. 7). Future efforts on blocking KIM1-DR5 interaction could employ additional tools like antibodies and small molecule drugs.

**Fig. 4 | KIM1 binds DR5 and activates its downstream caspase cascade. a** Co-IP of KIM1 and DR5 in HK-2 cells transfected with the plasmids indicated. The Co-IP assay was performed using anti-HA or respective IgG, while IgG served as the negative control. For IP and lysate groups, HA-tagged KIM1 was detected with anti-HA antibody, and Flag-DR5 was detected using anti-DR5 antibody. **b** Co-IP of KIM1 and DR5 with/without 24 h cisplatin (Cis) in HK-2 cells. The Co-IP assay used anti-KIM1 or respective IgG. For IP and lysate groups, KIM1 was detected with anti-KIM1 antibody, and DR5 was detected using anti-DR5 antibody. **a, b** Each experiment was repeated at least three times independently with similar results obtained. **c** Schematic diagram for FRET assay design. The excitation/emission wavelength of CFP and YFP are 435/485 nm and 485/525 nm respectively. Emission of CFP excites YFP that causes FRET detected at 525 nm; FRET signal is detectable when KIM1-CFP binds DR5-YFP (<10 nm). **d, e** Quantitative FRET signal between KIM1-CFP and DR5-YFP with/without 24 h cisplatin using fluorescence scan (**d**), and wavelength scan (**e**). **d** CFP, pRK-5′Flag-CFP; YFP, pRK-5′Flag-YFP; KIM1-CFP, pRK-5′Flag-KIM1-CFP; DR5-YFP, pRK-5′Flag-KIM1-YFP; *n* = 3 biological samples per group, each experiment was repeated at least three times independently with similar results obtained. **e** Each experiment was repeated at least three times independently with similar results obtained. **f** Representative images of immunohistochemical staining and quantitative results of DR5 on mouse renal sections at Day 3 after cisplatin injury or unilateral renal ischemia-reperfusion injury (uIRI). Scale bar, 50 μm. *n* = 4 mice per group. **g, h** The effects of KIM1 overexpression (**g**) and knockout (**h**) on activation of DR5 downstream caspase cascade after 24 h cisplatin injury in HK-2 cells. KIM1 KO, lenti-CRISPR/Cas9-based KIM1 knockout. **g, h** Each experiment was repeated at least three times independently with similar results obtained. **i, j** Representative images of KIM1 and DR5 staining in HK-2 cells with/without 24 h cisplatin (**i**) and mouse renal sections at Day 3 after cisplatin injury (**j**). Scale bars, 50 μm. **i** *n* = 3 biological samples per group; **j** *n* = 4 mice per group. **k–m** MTT assay (**k**), mRNA levels of apoptotic molecules (**l**), and DR5 downstream caspase cascade (**m**) in HK-2 cells, with KIM1 overexpression and/or DR5 knockdown, with/without 24 h cisplatin. **k** *n* = 5 biological samples per group; **l** Vec, *n* = 3 biological samples per group. Each experiment was repeated at least three times independently with similar results obtained. **m** Each experiment was repeated at least three times independently with similar results obtained. pRK-5′Flag; KIM1, pRK-5′Flag-KIM1; pSuper, pSuper backbone; ShDR5, pSuper-ShDR5; CT, without cisplatin treatment; data shown as mean ± SD. Two-tailed unpaired Student's *t*-test was used for two experimental groups, and one-way ANOVA for multiple experimental groups without adjustment. *$P < 0.05$; **$P < 0.01$; ns no significance. Exact *P* values are provided in Source Data.

In this work, we report that renal tubular specific *Kim1* knockout relieves cisplatin and IR-injured AKI, unveiling a role for the YY1-KIM1-DR5 axis in the progression of AKI, our study also provides a proof-of-concept into AKI treatment through targeting the YY1-KIM1-DR5 axis (Fig. 8l).

## Methods

### Study approval
Animals were handled according to the Guidelines of the China Animal Welfare Legislation, as approved by the Committee on Ethics in the Care and Use of Laboratory Animals of College of Life Sciences, Wuhan University.

### Animals, AKI models and treatments
*Kim1*^flox/flox^ mice in C57BL/6 background were generated by Gempharmatech (Nanjing, China), which were crossed with *Ksp-Cre* mice (CDH16-Cre, Cre activity expressing in renal tubules; kind gifts from Dr. Congyi Wang, Huazhong University of Science and Technology) to generate renal tubular specific *Kim1* knockout (*Kim1*^*Ksp*^ KO) mice. Age-matched (8–12 weeks old, 25 ± 3 g) male wildtype (WT) and *Kim1*^*Ksp*^ KO littermates were used. Male C57BL/6 mice (8–12 weeks old, 25 ± 3 g) were obtained from Hubei Center for Disease Control and Prevention. Mice were maintained in a specific-pathogen-free, temperature controlled (22 ± 1 °C) animal facility with a 12 h light/dark cycle, and free access to water/food. Male mice were used in this study, euthanasia was performed by $CO_2$ inhalation. Mice were maintained on normal chow (NC, #1025, Beijing Huafukang, Beijing, China).

Both unilateral and bilateral IRI-induced AKI models were used[40–42]. Mice were anesthetized and underwent midline abdominal incisions. For uIRI, an AKI model that exhibits lower mortality but is less suitable for evaluating renal function since only one kidney is injured, the left renal pedicle was clamped for 45 min. For bIRI, an AKI model with higher mortality but more accurate renal function evaluation, both kidneys were clamped to block blood flow for 30 min. Mice were euthanized at Days 1, 3, 7 and 21 after uIRI, and at Day 1 after bIRI, and blood and kidneys were collected.

For cisplatin-induced AKI, a single dose of 30 mg/kg body-weight (BW) cisplatin was injected intraperitoneally and the mice were euthanized at Day 1 or 3 after the injection[43]. For treatments, in cisplatin-injured mice, different dosages of P2 peptide (0.2, 1, 2 or 5 μM) dissolved in PBS containing 1% DMSO were intravenously injected, or eudesmin (10 mg/kg BW) was intraperitoneally injected, with the first injection at 30 min after cisplatin injury, then twice per day for 3 days.

### Chemicals and antibodies
Cisplatin (T1564), eudesmin (T3836) and nystatin (T1678) were from TargetMol Chemicals (Shanghai, China). FITC (F7250) was obtained from Sigma (Saint Louis, MO). 5′(6)-FAM (HY1537) was obtained from MCE (New Jersey, NY). Cy7 NHS ester (25020) was obtained from Lumiprobe (New Jersey, NY). The antibody against KIM1 (dilution 1:1000 for WB, 1:200 for immunohistochemistry and immunofluorescence) was prepared by DaiAn Biotech (Wuhan, China), using the antigenic determinant (CQNGIVWTNGTHVTYRKDTR) as selected by Antibody Epitope Prediction (http://tools.immuneepitope.org/bcell) from the Ig V domain of KIM1, and was used for Western blotting and immunostaining. Antibodies against β-actin (dilution 1:10000, A5316, clone number AC-74), Flag (dilution 1:5000, F1804, clone number M2) and HA (dilution 1:5000, H3663, clone number HA-7) were from Sigma Aldrich. Antibodies against c-Caspase3 (dilution 1:1000, 9661), DR5 (dilution 1:1000 for WB, 1:200 for immunohistochemistry and immunofluorescence, 8074, clone number D4E9), PARP-1 (dilution 1:1000, 9532, clone number 46D11), p-p53 (dilution 1:1000, 9284), F4/80 (dilution 1:100, 70076, clone number D2S9R) and YY1 (dilution 1:1000 for WB, 1:200 for immunohistochemistry, 46395, clone number D5D9Z) were from Cell Signaling Technology (Boston, MA). Antibody against Caspase3 (dilution 1:1000, A2156) was from Abclonal (Wuhan, China). The antibody against DR5 (dilution 1:200 for immunofluorescence, Sc-166624, clone number D-6) used for immunofluorescence was from Santa Cruz (Santa Cruz, CA). Antibodies against Caspase8 (dilution 1:1000, 13423-1-AP) and TRAIL (dilution 1:1000, 27064-1-AP) were from Proteintech (Chicago, IL). Antibodies against Caspase9 (dilution 1:1000, ab202068, clone number EPR18107) and Hsp70 (dilution 1:5000, 610607) were from Abcam (Cambridge, UK) and BD (New Jersey, NY), respectively. All antibodies used in this study are from commercial suppliers who have verified the specificity of the antibodies by using recombinant proteins or knockout cell lines.

### Cell culture, mPTEC cells isolation, and treatments
The human kidney tubular cell line HK-2 (GDC0152, from China Center for Type Culture Collection, CCTCC, Wuhan, China) was cultured in DMEM/F12 medium (Hyclone, Logan, UT) with 10% fetal bovine serum. The mouse kidney tubular cell line TCMK-1 (CCL-139, from National Collection of Authenticated Cell Cultures, Shanghai, China) and human embryonic kidney cell line HEK293T (CL-0005, from Procell Biotech, Wuhan, China) were cultured in DMEM/High glucose (Hyclone) and 10% fetal bovine serum.

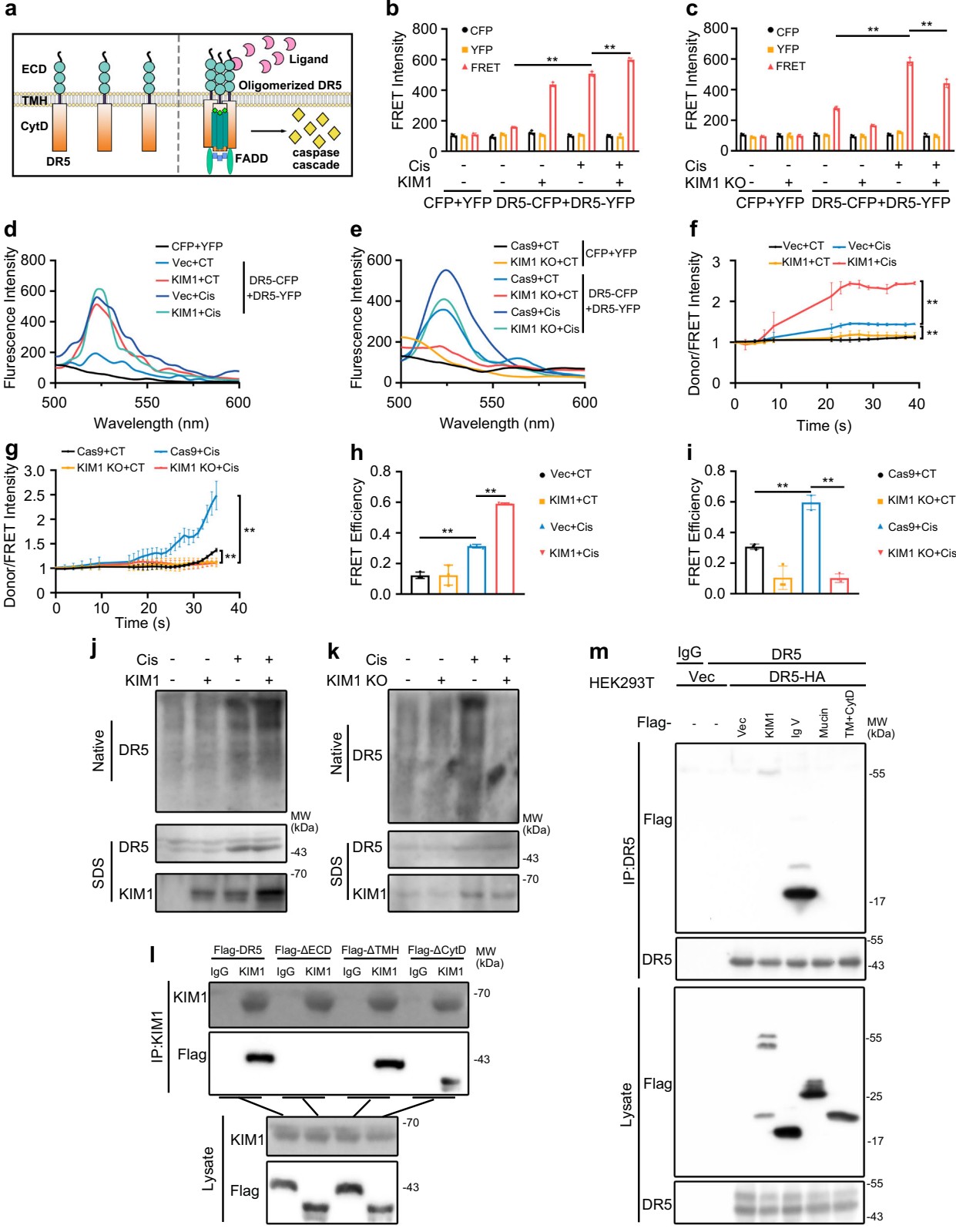

Mouse primary tubular epithelial cells (mPTEC) were isolated[44]. Briefly, minced renal cortex was digested in 1 mg/ml type II collagenase (Sigma, Saint Louis, MO), and sequentially passed through 200 μm and 70 μm cell strainers. Cells were cultured in RPMI-1640 medium (Hyclone) with 5 mg/L epidermal growth factor (PeproTech, Rocky Hill, NJ).

For treatment with peptides P1, P2 and P3, HK-2 or TCMK-1 cells were co-treated with 5 μg/mL cisplatin dissolved in PBS containing 0.1% DMSO and 1 μM or 10 μM of the respective peptide for 24 h. For

eudesmin treatment, HK-2 cells were co-treated with 5 μg/mL cisplatin and 1 μM eudesmin. For nystatin treatments, HK-2 cells or mPTECs were co-treated with 5 μg/mL cisplatin and 10 μg/mL, 50 μg/mL or 100 μg/mL nystatin.

**Biochemical assessment of serum**

Mouse renal function was evaluated by serum creatinine and BUN levels using a creatinine or a BUN reagent kit (Jiancheng Bio., Nanjing,

**Fig. 5 | KIM1 promotes the multimerization of DR5. a** Structural scheme of DR5 multimerization and downstream signaling pathways. Under physiological conditions, DR5 exists as a monomer, upon injury, ECD binds ligand and multimerizes, subsequently recruiting FADD with its CytD and activating the downstream caspase cascade. ECD ectodomain, CytD cytoplasmic domain. **b–e** Quantitative FRET indicating the effect of KIM overexpression (**b**) or knockdown (**c**) on DR5 multimerization following cisplatin injury, using fluorescence (**d**) or wavelength (**e**) scan. CFP, pRK-5′Flag-CFP; YFP, pRK-5′Flag-YFP; DR5-CFP, pRK-5′Flag-DR5-CFP; DR5-YFP, pRK-5′Flag-DR5-YFP; Vec, pRK-5′Flag; KIM1, pRK-5′Flag-KIM1; Cas9, lenti-CRISPR/ Cas9; KIM1 KO, lenti-CRISPR/Cas9-based KIM1 knockout. **b, c** n = 3 biological samples per group, each experiment was repeated at least three times independently with similar results obtained; **d, e** Each experiment was repeated at least three times independently with similar results obtained. **f, g** Fluorescence redistribution after photobleaching (FRAP) assays showed the effect of KIM1 overexpression (**f**) or knockdown (**g**) on DR5 multimerization. **f, g** n = 3 biological samples per group, each experiment was repeated at least three times independently with similar results obtained. **h, i** FRET efficiency in KIM1 overexpression (**h**) or knockdown (**i**) groups. **h, i** n = 3 biological samples per group, each experiment was repeated at least three times independently with similar results obtained. **j, k** The effect of KIM1 overexpression (**j**) or knockdown (**k**) on the formation of higher-order DR5 oligomers detected using native PAGE electrophoresis. Each experiment was repeated at least three times independently with similar results obtained. **l** KIM1 bound to the ECD domain of DR5. A Co-IP assay was performed using anti-KIM1 or non-specific IgG as the negative control. For IP and lysate groups, KIM1 was detected with anti-KIM1 antibody; WT and mutants of DR5-Flag were detected using an anti-Flag antibody. **m** DR5 bound to the Ig V domain of KIM1. Co-IP assay was performed using anti-KIM1 or respective IgG, while IgG served as the negative control. For IP and lysate groups, Flag-tagged KIM1 truncations were detected with an anti-Flag antibody, and DR5-HA was detected using an anti-DR5 antibody. **l, m** Each experiment was repeated at least three times independently with similar results obtained. Data are shown as mean ± SD. Two-tailed unpaired Student's t-test was used for two experimental groups, and one-way ANOVA for multiple experimental groups without adjustment. **P < 0.01. Exact P values are provided in Source Data.

China)[45]. Mouse serum IL-6 was detected using an ELISA kit according to the manufactory instructions (Absin, Shanghai, China).

### Ex vivo imaging of peptide P2
For ex vivo imaging, P2 was labeled with Cy7, a near-infrared fluorescein. Cisplatin (30 mg/kg BW) was intraperitoneally injected into 8–12 weeks old male mice. 24 h later, Cy7-labeled P2 (1 μM) was intravenously injected. 4 h later, mice were sacrificed and major organs including brain, heart, lung, liver, spleen and kidney were imaged ex vivo using Trilogy digital radiography (LI-COR, Troy, MI).

### Transcriptional factors prediction and GTRD analysis
Transcription factors that bind to the *KIM1* promoter were predicted using JASPAR (https://jaspar.genereg.net), Human TFBD 3.0 (http:// bioinfo.life.hust.edu.cn/ Human TFDB) and hTFtarget (http://bioinfo. life.hust.edu.cn/hTFtarget)[46–48]. Intersecting transcription factors that potentially regulate KIM1 expression were further screened. GTRD (Gene Transcription Regulation Database) (http://gtrd.biouml.org), which provides transcription factor binding sites analysis based on ChIP-Seq, was used to analyze YY1-binding sites on *KIM1* promoter[19].

### Constructs and transfections
pECFP (P0374) and pEYFP (P5328) were from Miaoling Bio (Wuhan, China). Plasmids expressing Flag-tagged human KIM1, KIM1 Ig V, KIM1 Mucin, KIM1 CytD, DR5, DR5 ΔECD, DR5 ΔTMH, DR5 ΔCytD; DR5-CFP, DR5-YFP, STAT1 and STAT3; as well as HA-tagged KIM1, DR5 and YY1, were constructed. shYY1 (target sequence 5′-3′: GGCGACGACGACTACA TTGAA) and shDR5 (target sequence 5′-3′: GGTAGAGATTGCAT CTCCTGC) cDNA were obtained from Tsing Biotech (Shanghai, China) and cloned into pSuper backbone, the KIM1 promoter was cloned into pGL3-enhancer backbone. CRISPR-Cas9-based knockout was performed as previously reported (guide RNAs, forward: CACCGCTGAC GGCCAATACCACTAA; reverse: AAACTTAGTGGTATTGGCCGTCAGC)[43]. Transfections were performed using Lipofectamine 2000 (Invitrogen, Carlsbad, CA).

### Cell death detection assays
TUNEL assay (Beyotime, Shanghai, China) was performed to detect in situ cell death of HK-2 cells or mouse renal sections and quantitated[44]. For apoptosis, cells that had been subjected to different treatments were labeled with Annexin V-FITC and propidium iodide (PI) with an apoptosis detection kit (BB-412, Best Bio., Shanghai, China); apoptotic cells were detected using a BD FACSAria flow cytometer (Franklin Lakes, NJ) and analyzed by FlowJo V10 (BD). Quantitative data provided the average and standard deviation from three independent experiments (percentage of apoptotic cells was calculated by Annexin-V positive cells (Q2 + Q3)).

### Native PAGE electrophoresis
Native PAGE was performed to evaluate DR5 multimerization. Cells or renal tissues were lysed in NETN buffer (20 mM Tris-HCl (pH 8.0), 100 mM NaCl, 0.5 mM EDTA, 0.5% NP-40) supplemented with protease inhibitor (HY-K0023, MCE) and 0.1 mM PMSF (Sigma), using a freeze-thaw cycle 3 times. Protein concentration was determined, an equivalent amount of protein from each sample was mixed with loading buffer free of SDS/DTT/β-mercaptoethanol, protein complexes were resolved by 12% native PAGE, transferred onto PVDF membrane and detected using an anti-DR5 antibody.

### Fluorescence resonance energy transfer (FRET) assay
Multimerization of DR5 was detected by FRET assay[49]. Briefly, plasmids expressing DR5-CFP and DR5-YFP were co-transfected into HK-2 cells, FRET signals were detected using a Hitachi F-2700 fluorescence spectrophotometer (Tokyo, Japan), and a Nikon AX confocal microscope (Tokyo, Japan). For spectrophotometer-based detection, cells were lysed at 24 h after transfection and detected by wavelength scan (500–600 nm) and time scan (435/525 nm, excitation/emission). FRET intensity was assessed using spectrophotometer-based detection, which records the FRET signal by the excitation of CFP and emission of YFP. Strong FRET intensity indicates strong interaction between detected molecules. For confocal-based detection, cells were imaged using the channels indicated (CFP: 435/485 nm; YFP: 485/ 525 nm; FRET: 435/525 nm), co-transfection of CFP and YFP plasmids was used as a control[50]. For acceptor photobleaching based FRET assays, YFP fluorescence was bleached to 50% of its initial intensity. Confocal images and lifetime imaging were acquired and FRET efficiencies were calculated by the intensity of donor before (Ipre) and after (Ipost) the photobleaching with the formula: $E_{FRET} = (1 - Ipre/Ipost)$[51,52].

### Co-immunoprecipitation
Cells were lysed in pre-lysis/wash buffer (25 mM Tris-HCl, pH 7.4, 150 mM NaCl, 1% NP-40, 1 mM EDTA, 5% glycerol); the cell lysate was sonicated and centrifuged at 12000 g for 10 mins. For IP, cell lysates were immunoprecipitated with the indicated antibody or respective IgG with Protein G magnetic beads (Bio-Rad, Hercules, CA) overnight at 4 °C[53]. After washing, the beads were boiled in loading buffer and subjected to immunoblotting.

### Identification of KIM1 binding partners
As we previously reported[54], KIM1 binding partners were immunoprecipitated by Flag antibody in HEK293T cells transfected with Flag-tagged KIM1. Altered bands in SDS-PAGE were digested in-gel and analyzed using a Q Exactive HF mass-spectrometer coupled with Easy-nLC 1000 system (Thermo Scientific, Rockford, IL).

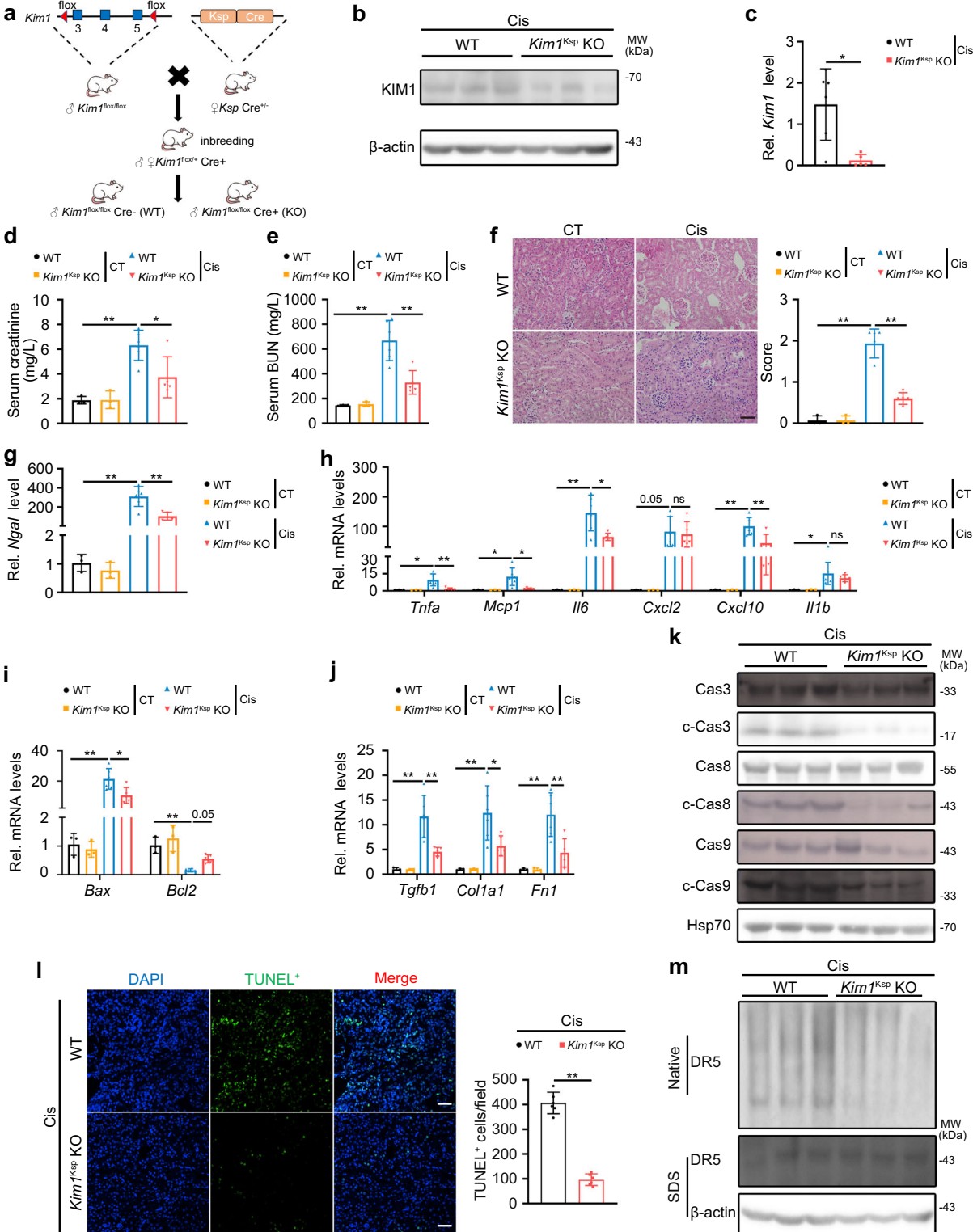

**Fig. 6 | Renal tubular specific knockout of *Kim1* relieves cisplatin-induced AKI.**
**a** Generation of renal tubular specific *Kim1* knockout mouse. Wildtype, WT, *Kim1*flox/flox; renal tubular specific *Kim1* knockout, *Kim1*Ksp KO. **b, c** KIM1 protein levels (**b**) and mRNA levels (**c**) in the kidneys of WT and *Kim1*Ksp KO mice at Day 3 after cisplatin injury (Cis). **b** n = 3 mice per group; **c** n = 6 for WT group and n = 5 for *Kim1*Ksp KO group, respectively. **d–f** Serum creatinine level (**d**), serum urea nitrogen level (**e**), and pathological score (**f**) of WT and *Kim1*Ksp KO mice at Day 3 after cisplatin injury. Scale bar, 50 μm. **d–f** n = 3 for WT and *Kim1*Ksp KO in CT groups; n = 6 for WT and n = 5 for *Kim1*Ksp KO in Cis-injured groups, respectively. **g–j** qPCR analysis of *Ngal* (**g**), inflammatory factors (**h**), apoptotic molecules (**i**) and fibrotic factors (**j**) in the kidneys of WT and *Kim1*Ksp KO mice at Day 3 after cisplatin injury.

**g–j** n = 3 for WT and *Kim1*Ksp KO in CT groups; n = 6 for WT and n = 5 for *Kim1*Ksp KO in Cis-injured groups, respectively. **k** Western blots of caspase cascade proteins from cisplatin-injured kidneys of WT and *Kim1*Ksp KO mice. n = 3 mice per group. **l** Representative images for TUNEL assay with quantitative results of cisplatin-injured WT and *Kim1*Ksp KO mice. Scale bar, 50 μm. n = 6 mice per group. **m** Native PAGE electrophoresis of DR5 oligomers in cisplatin-injured WT and *Kim1*Ksp KO mice. n = 3 mice per group. Data shown as mean ± SD. Two-tailed unpaired Student's t-test was used for two experimental groups, and one-way ANOVA for multiple experimental groups without adjustment. *P < 0.05; **P < 0.01; ns no significance. Exact P values are provided in Source Data.

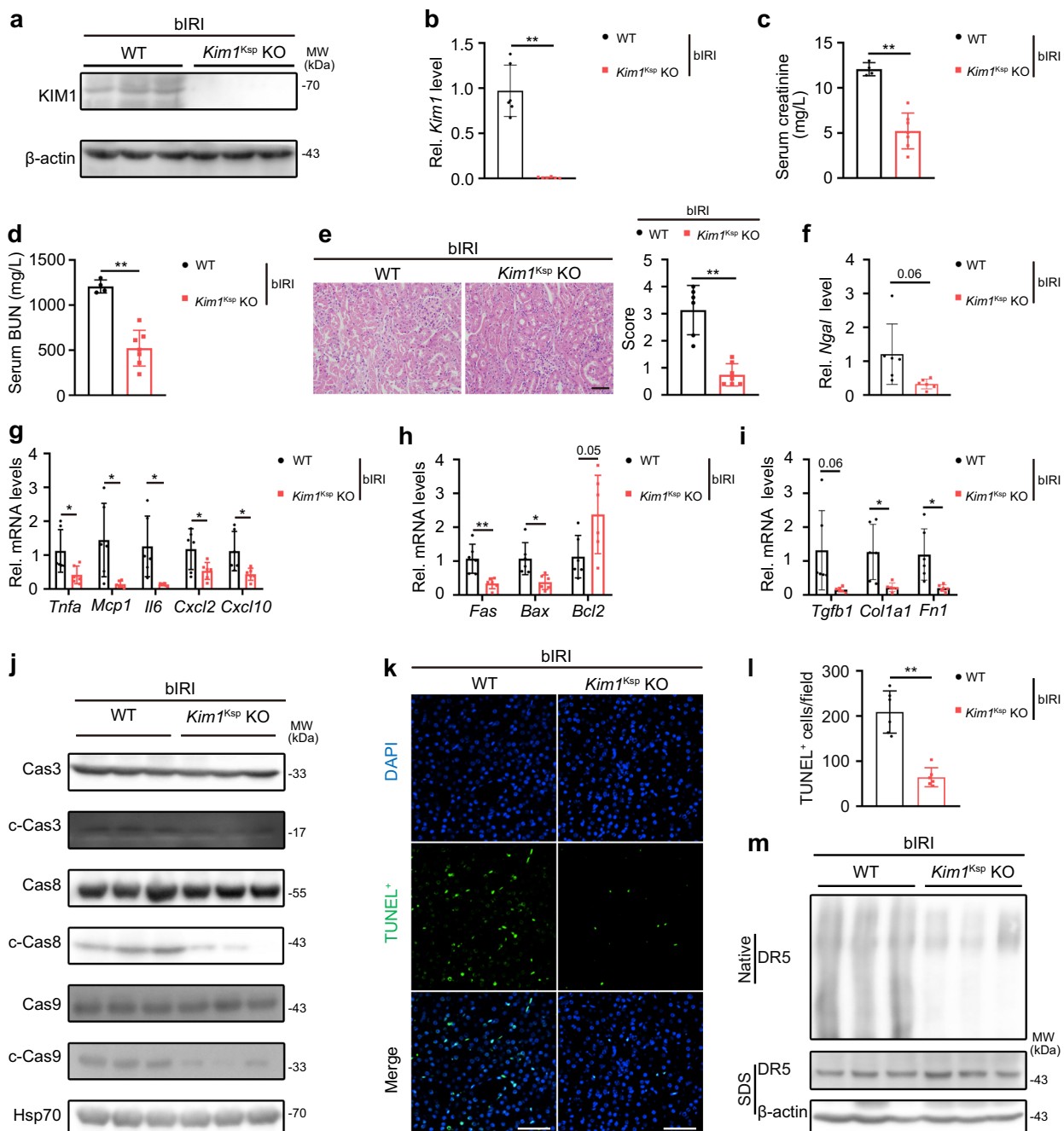

**Fig. 7 | Renal tubular specific knockout *Kim1* relieves bilateral ischemia-reperfusion-induced AKI. a, b** Western blots (**a**) and qPCR (**b**) of KIM1 levels in the kidneys of WT and *Kim1*[Ksp] KO mice at Day 1 after bilateral renal ischemia-reperfusion injury (bIRI). **a** n = 3 mice per group; **b** n = 6 mice per group. **c, e** Serum creatinine level (**c**), serum urea nitrogen level (**d**), and pathological score (**e**) of WT and *Kim1*[Ksp] KO mice at Day 1 after bIRI. Scale bar, 50 μm. **c, d** n = 4 for WT group and n = 7 for *Kim1*[Ksp] KO group, respectively. **e** n = 6 for WT group and n = 7 for *Kim1*[Ksp] KO group, respectively. **f–i** qPCR of *Ngal* (**f**), inflammatory factors (**g**), apoptotic molecules (**h**) and fibrotic factors (**i**) in the kidneys of WT and *Kim1*[Ksp] KO mice at Day 1 after bIRI. **f, i** n = 6 mice group. **j** Western blots of caspase cascade proteins from the kidneys of WT and *Kim1*[Ksp] KO mice at Day 1 after bIRI. n = 3 mice per group. **k, l** Representative images of TUNEL assay (**k**) with quantitative results (**l**) of WT and *Kim1*[Ksp] KO mice at Day 1 after bIRI. Scale bar, 50 μm. **k, l** n = 6 mice per group. **m** Native PAGE electrophoresis of DR5 oligomers in WT and *Kim1*[Ksp] KO mice at Day 1 after bIRI. n = 3 mice group. Data shown as mean ± SD. Two-tailed unpaired Student's t-test was used for two experimental groups, and one-way ANOVA for multiple experimental groups without adjustment. *P < 0.05; **P < 0.01. Exact P values are provided in Source Data.

## Chromatin immunoprecipitation (ChIP)

Cells were crosslinked with 1% formaldehyde then quenched with glycine (125 mM). After washing with PBS, samples were resuspended with digestion buffer plus 1 mM PMSF[55,56]. Chromatin was then sonicated and immunoprecipitated using anti-YY1 antibody or rabbit IgG (Sigma). Purified DNA was detected by qPCR (primers listed in Supplementary Table 6), with the inputs as the internal control.

## Quantitative real-time PCR (qPCR)

Cells were treated with cisplatin (5 μg/mL), etoposide (50 μM, T0132, TargetMol), human IL-6 (Peprotech, 50 ng/mL), human TNF-α (Peprotech, 20 ng/mL), $H_2O_2$ (800 μM, Sinopharm, Beijing, China) for the time indicated, total RNA was isolated from cells using RNA[iso] Plus (TaKaRa, Dalian, China) and subjected to qPCR analysis[57]. The mRNA levels of specific genes were normalized to β-actin. Primers used for qPCR are provided in Supplementary Table 7.

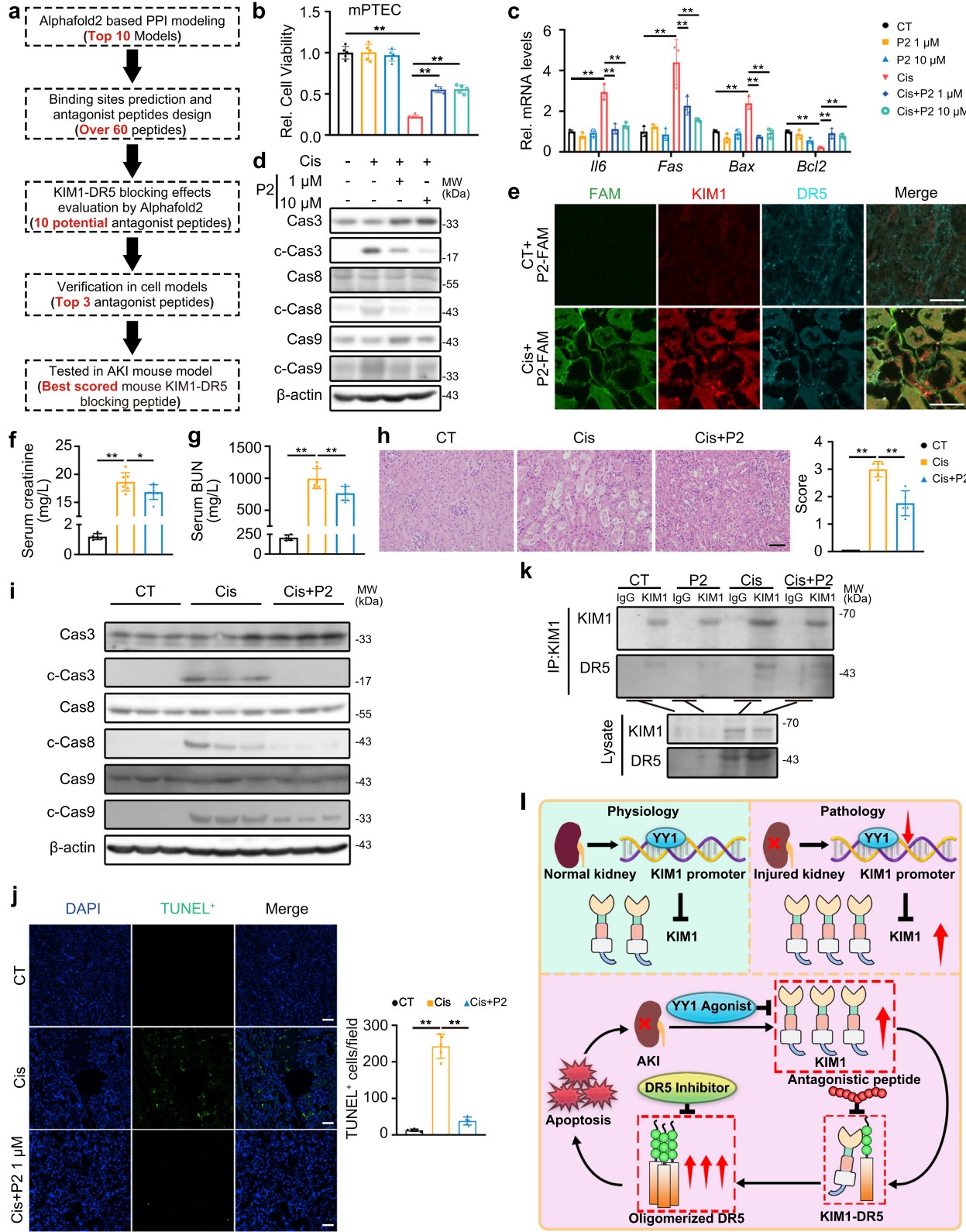

### Luciferase reporter assay

Regions from -2000 to 0 relative to transcription start site (TSS) of the *KIM1* promoter were cloned into pGL3-enhancer vector. HEK293T cells were transfected with indicated plasmids for 24 h and luciferase assays were performed and analyzed[54].

### Cell viability assay

HK-2, TCMK-1 and mPTECs cells were plated in 96-well plates and grown to 80% confluence. The cells were incubated with 5 μg/mL cisplatin dissolved in PBS containing 0.1% DMSO with or without different treatments for 24 h. 10 μL MTT (5 mg/mL) was added to each well for

**Fig. 8 | Rationally designed antagonistic peptides block KIM1-DR5 interaction and protect against AKI. a** Workflow for the screen and evaluation of antagonistic peptides blocking KIM1-DR5 interaction. **b** MTT assays showed the protective effects of peptide P2 in 24 h cisplatin-injured mPTECs (mouse Primary Renal Tubular Epithelial Cells). $n = 5$ biological samples per group, each experiment was repeated at least three times independently with similar results obtained. **c, d** qPCR analysis of apoptotic molecules (**c**) and caspase cascade activation (**d**) in 24 h cisplatin-injured TCMK-1 cells treated with/without peptide P2. **c** $n = 3$ biological samples per group, each experiment was repeated at least three times independently with similar results obtained; **d** Each experiment was repeated at least three times independently with similar results obtained. **e** Representative images of P2-5'(6)-FAM (green), KIM1(red) and DR5 (cyan) on mouse renal sections at Day 3 after cisplatin injury. Scale bar, 50 μm. $n = 3$ mice per group. **f, g** Serum creatinine (**f**) and urea nitrogen levels (**g**) of peptide P2 treated mice at Day 3 after cisplatin injury. **f, g** $n = 5$ for CT group and $n = 7$ for Cis-injured and Cis+P2 groups, respectively. **h** Representative H&E staining and pathological score of P2 treated mice at Day 3 after cisplatin injury. Scale bar, 50 μm. $n = 5$ mice per group. **i** Western blots of

caspase cascade of peptide P2 treated mice at Day 3 after cisplatin injury. $n = 3$ mice per group. **j** Representative images of TUNEL assay with quantitative results on renal sections of peptide P2 treated mice at Day 3 after cisplatin injury. $n = 5$ mice per group. **k** Co-immunoprecipitation showed reduced endogenous KIM1-DR5 interaction in the kidneys of peptide P2 treated mice at Day 3 after cisplatin injury. Co-IP was performed using anti-KIM1 or respective IgG, while IgG served as the negative control. For IP and lysate groups, KIM1 was detected using an anti-KIM1 antibody, and DR5 was detected using an anti-DR5 antibody. Pooled samples from 4 mice were used for each lane. **l** Working model of YY1-KIM1-DR5 axis. Under physiological conditions, YY1 binds to the promoter of *KIM1* and represses its expression. In injured kidney, downregulated YY1 up-regulates *KIM1*, which binds DR5 and promotes its multimerization, activates the downstream caspase cascade, leads to apoptosis and aggravates AKI. Data shown as mean ± SD. Two-tailed unpaired Student's *t*-test was used for two experimental groups, and one-way ANOVA for multiple experimental groups without adjustment. $^{*}P < 0.05$; $^{**}P < 0.01$. Exact *P* values are provided in Source Data.

another 4 h. Medium was then removed and DMSO was added. Absorbance was measured at 490 nm and normalized to the control group.

## Histological and immunohistochemical studies
Renal sections were stained with hematoxylin and eosin, and the degree of renal damage was determined in a double-blind manner[40]. 1 μg/mL PNA (peanut agglutinin) or LTL (lotus tetragonolobus lectin; both from Vector Laboratories, Burlingham, CA) were used to detect renal distal tubules or proximal tubules, respectively. For KIM1, DR5 and F4/80 immunostaining or immunohistochemistry staining, renal sections or HK-2 cells were incubated overnight at 4 °C with the respective primary and secondary antibodies (Thermo Fisher, Waltham, MA). For P2/KIM1/DR5 triple staining, cisplatin-treated HK-2 cells or control cells were incubated with FITC-labeled P2 for 4 h; Day 1 cisplatin-injured and uninjured mice were injected intravenously with 5'(6)-FAM-labeled P2 for 4 h; P2 was detected by labeling with fluorescein while KIM1/DR5 was detected by immunostaining. Sections or cells were then stained with 4,6-diamidino-2-phenylindole (DAPI) dye and antifading medium. Images were taken using a confocal microscope (Nikon AX, Japan) with at least 5 fields per sample. Relative integrated option density (IOD) was calculated using Image J 1.53[58].

## PPI modeling, design and evaluation of KIM1-DR5 blocking peptides
KIM1-DR5 interaction was modeled by Alphafold2 repository, with each of the five trained model parameters. Query sequence and MSA from MMseqs251 were input without templates[28]. MSA generation and AlphaFold2 predictions were performed using ColabFold (https://colab.research.google.com/ssssub/sokrypton/ColabFold). Modeling results were analyzed using PDBePISA (www.ebi.ac.uk/msd-srv/prot_int/pistart.html). The binding sites of KIM1 and DR5 were analyzed by $\Delta^i G$ (solvation energies) and RMSD. Protein-protein interaction was visualized with Pymol 2.3.0. Hot spots that contribute to PPI were selected and visualized; candidate peptides that block KIM1-DR5 interaction were designed based on the hot spots identified (Supplementary Table 4). Candidate peptides were chemically synthesized by GenScript (Nanjing, China) and their bioactivities were evaluated in cisplatin-injured cells or mice.

## Statistical analyses
Data were analyzed with GraphPad Prism 8.0. Statistical analysis was performed using a two-tailed Student's *t*-test for two experimental groups, and one-way ANOVA for multiple experimental groups without adjustment. Tukey's multiple-comparison test was used as post hoc test to identify between-group differences. Data are reported as the mean values with error bars showing the standard deviation (SD). $P < 0.05$ was considered statistically significant.

## Reporting summary
Further information on research design is available in the Nature Portfolio Reporting Summary linked to this article.

## Data availability
All data relating to this study can be found in the main text, figures or supplementary information. Source data are provided with this paper. Databases used in this study include Human Protein Atlas (https://www.proteinatlas.org), JASPAR (https://jaspar.genereg.net), hTFtarget (http://bioinfo.life.hust.edu.cn/hTFtarget), Human TFBD 3.0 (http://bioinfo.life.hust.edu.cn/HumanTFDB) and GTRD (http://gtrd.biouml.org). Source data are provided with this paper.

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

## Acknowledgements

This work is supported by the National Key R&D Program of China (2022YFA0806100 to K.H.; 2018YFA0800700 and 2019YFA0802701 to L.Z.), the Natural Science Foundation of China (32021003 to L.Z.; 31971066 and 82273838 to K.H.), the Fundamental Research Funds for the Central Universities (2042022dx0003 to L.Z.), and the Natural Science Foundation of Hubei Province (2021CFA004 to K.H.; 2022CFB247 to H.C., and 2022DFE025 to A.P.). The work was technically supported by the Analytical and Testing Center of Huazhong University of Science and Technology.

## Author contributions

C.Y., H.C., L.Z. and K.H. designed the study and analyzed the data. C.Y., H.X., M.X., D.Y., Y.F., X.L., Y.Zhang, Y.Xie, Y.C., H.C., A.P., Y.Xiao and Y.Zhou performed the experiments. Y.Xiao, L.S. and C.W. performed the molecular simulations. C.Y., H.X., R.B.P., H.C., L.Z. and K.H. wrote the manuscript.

## Competing interests

The authors declare no competing interests.
