## [Peer Review File · Nature Communications]

A renal YY1-KIM1-DR5 axis regulates the progression of acute kidney injuryREVIEWER COMMENTS

Reviewer #1 (Remarks to the Author):

This is a very interesting study which studied the YY1-KIM1-DR5 axis in the progression of AKI. The most important finding of this study is that YY1 transcriptionally regulates KIM1 and KIM1 interacts with DR5. This axis was demonstrated to be involved in the progression of AKI.

1. The downstream of KIM1 in this study has been clearly elaborated in this study, including those downstream pathways and phenotypic alterations. However, the mechanism of YY1 upregulation was less-well studied in the current study. As in the current models, both irradiation and cisplatin induced DNA damage, could DNA damage be a driving factor for YY1 upregulation.
2. Since the YY1 inhibitor is available, it is very interesting to see if the YY1 inhibitor could reverse or alleviate the progression of AKI
3. YY1 has been extensively studied in many diseases, could the authors use the CHIP-seq data from other studies to study if the KIM1 was a potential target of YY1
4. The methodology is very sound, and enough details were provided, and I have no further suggestions for this part.

Reviewer #2 (Remarks to the Author):

The manuscript investigated the regulation of kim-1 in the progression of acute kidney injury.

Comments

- 1) KIM-1 has been well investigated and validated as a marker of AKI. Both models of AKI, cisplatin and ischemia-reperfusion AKI were applied. Could authors explain any difference of the two models in the expression pattern/kinetic of KIM-1?
- 2) It has been reported that KIM-1 levels increased after one-day post-cisplatin administration and this elevation correlated with AKI (Int. J. Mol. Sci. 2019, 20(20), 5238); Moreover, KIM-1 expression is at peak 24 h after IRI (doi: 10.7150/thno.73426). In the present study, for cisplatin model was intraperitoneally injected into 8-week-old male mice and euthanized on day 3 after the 5 injections; for IRI, mice were euthanized at indicated day after injury to harvest blood and kidneys, but the time points were not well mentioned. The time points should be well selected to investigate the expression of KIM-1;
- 3) Transcription factor Yin Yang 1 (YY1), downregulated upon AKI, bound to the promoter of KIM1 and repressed its expression. Again, the expression pattern of YY1 should be further investigated. Is YY1 expression ahead of KIM-1?
- 4) In present study, authors draw the conclusion that KIM1 aggravates inflammation and apoptosis in renal tubular epithelial cells. A previous study (J Clin Invest. 2015; 125(4): 1620-36) reported KIM-1 protects the kidney after acute injury by downregulating innate immunity and inflammation. Only in vitro cell experiments were done in the present study to confirm this and further animal experiments should be done;
- 5) The expression of KIM-1 in Fig 1C, D was not consistent;
- 6) The experimental groups should be well labeled and noted in figure legends, e.g., CT, CT+Ctrl, Kim1^{Δksp} KO (transgenic model only or with AKI, time points), et al.

Reviewer #3 (Remarks to the Author):

The paper "A renal YY1-KIM1-DR5 axis regulates the progression of acute kidney injury" by Chen Yang et al. investigates the role of the interaction between DR5 and KIM1 in acute kidney injury. Specific rationally designed peptides able to inhibit the interaction between DR5 and KIM1 were also able to reduce the renal toxicity and cytotoxicity induced by cisplatin treatment.

The topic is certainly important and deserves the attention of translational research. However, this work has many limitations. Although many different methodologies have been employed, in some

parts the results are unclear.

Meanwhile, it is not conclusive about the message. That is, is the interaction between DR5 and KIM1 causal or not with respect to the development of renal failure?

In my opinion, the work is missing a lot of important information that needs to be added:

1. Which alterations induces per se the overexpression of KIM1 in HK-2 cells such as to sensitize them to the treatment with cisplatin? For example, at the level of cytokine production, NF κ B (Nuclear Factor kappa B) activation, DR5 expression and/or dimerization, mitochondrial function, oxidative stress?
2. Is there a change in the molecular interaction between KIM1 and DR5 in KIM1 overexpressing cells?
3. In Materials and Methods section, the authors describe only the conditions related to FRET experiments to study the interaction between exogenous KIM1 (KIM1-HA) with overexpressed Flag-tagged DR5 in HK-2 cells. I have not found information on how the FRET experiments were carried out to study the interaction between endogenous KIM1 and DR5, which in my opinion are much more important. Were specific antibodies used? by FRET intensity do the authors mean the FRET efficiency?
4. Nystatin is not a DR5 dimerization inhibitor rather a lipid raft disorganizer. Since, as reported in previous papers, DR5 transmits the apoptotic signal only when localized in the rafts, the effect observed by the authors following pre-treatment with nystatin seems to depend on this aspect. Authors should try to use methyl- β -cyclodextrins or statins, which have the advantage of being less toxic than nystatin both in vitro and in vivo and perifosine, which, on the contrary, induced the recruitment of DR5 into lipid microdomains.
5. Many papers report an increase in apoptosis in AKI models both in vitro and in vivo. All that, even though different ways, can reduce apoptosis has been found to be effective in improving renal pathological injury. Among these: antioxidants and anti-inflammatories of various kinds, mitochondrial protectors, inhibitors of the opening of the mitochondrial pore, and pure inhibitors of apoptosis such as over-expression of Bcl-2. This further highlight that apoptosis represents a key event in the onset and evolution of the disease, at least that linked to drug toxicity, but equally underlines how a primum movens that could indicate a promising and innovative therapeutic target has not yet been identified.

Reviewer #4 (Remarks to the Author):

The authors describe the downregulation of YY1 upon Cisplatin or I/R-induced kidney injury to lead to an increase of KIM-1 in the kidney. In turn, upregulation of KIM-1 binds to DR-5, leading to gene expression of pro-inflammatory cytokines and induction of cell death. The authors convincingly demonstrate the effect of overexpression and knock-out of KIM-1 on relevant targets in the YY-1-KIM1-DR-5 pathway, as well as downstream inflammation and cell death. Experiments in Cisplatin-treated mice and cells and the induction of I/R in mice support the conclusion. The concise set of experiments seems to be well-designed and follows a clear strategy, using state-of-the-art methodology. The results sufficiently support the work's findings. The results are novel and noteworthy: KIM1 is a well-known biomarker for AKI and current results now demonstrate a mechanistic role of KIM1 in the pathogenesis of AKI, which are noteworthy results. However, I have identified several inconsistencies and textual issues that should be corrected prior to publication. Amongst others, additional data is required to understand the interaction sites between KIM1 and DR5, relate the protective effects of KIM1Ksp KO mice to the extent of AKI in WT mice with and without Cisplatin, and explain the protective effects of the P2 peptide. The method section lacks some details that preclude the reproduction of some of the experiments.

Major issues:

- The authors used different human kidney cell lines, namely HK-2, HEK-293, and 293T. What is the underlying reason, and to what extent may this influence the results?
- The authors developed an antagonistic peptide (P2) that blocked KIM1-DR5 interaction and exhibited reno-protective effects in vivo and in vitro. To interact with KIM1-DR5, the peptide needs to

be targeted to the intracellular milieu of tubular cells. It is unclear from the manuscript how the authors envision the peptide ending in the cells expressing KIM1-DR5. Can the authors demonstrate the localization of the P2 peptide into the target cells?

- The rationale for the dosage of P2 in mice is lacking. How did the authors decide on this dose? Was this extrapolated from the in vitro experiments? If so, how? P2 only partially prevented Cisplatin-induced cell death in the kidney, with only marginal effects on serum creatinine and BUN. Could this observation be due to relatively low tissue levels of P2? The manuscript could be improved by a) demonstrating levels of P2 in the kidney and b) demonstrating the effect of a higher dosage of P2 on the kidney upon Cisplatin treatment.
- The results from in silico mapping of the interaction sites (result: ECD deletion abolished interaction) and in vitro co-IP experiments (result: KIM1 binds to TMH) seem to contradict each other. How can the authors explain this?
- The authors demonstrate that KIM1 binds to the TMF-domain of DR5. However, it remains unclear which domain of KIM1 interacts with the TMH region in DR5. Can the authors demonstrate the binding domain of KIM1? The authors present a drawing on the interaction between KIM1-DR5 (Figure 7L) that does not seem to represent the findings in the manuscript. Although the authors demonstrate that KIM1 seems to bind TMH on DR5 (or ECG, based on in silico results?) they do not seem to provide evidence to support the binding domain of KIM1. Instead, the figure suggests that the cytosolic domain of KIM1 binds to the CytD domain of DR5, which partly contradicts their results and is also not fully supported by data. Although I am not sure whether the authors intended to demonstrate such a detailed view of the interactions between both proteins, I believe the manuscript could best be enriched with data revealing the interacting domains of KIM1 and DR5.
- The authors demonstrate lowered levels of serum creatinine, BUN, reduced gene expression of pro-inflammatory genes, and lower levels of cell death in KIM1Ksp KO animals treated with Cisplatin as compared to WT animals treated with Cisplatin (Figures 5 and 6). Yet, the authors do not demonstrate such values in WT animals not treated with Cisplatin, which makes it impossible to interpret the relevance of the findings beyond a proof-of-concept. To what extent does KIM1Ksp KO protect against AKI in Cisplatin-treated mice? The levels are lower than WT mice treated with Cisplatin, but how does this relate to healthy mice? The same issue applies to the lack of a control group for the I/R experiments in mice (Figure 6).
- The authors note that TRAIL can bind and activate DR5 in the discussion, and mention that the levels of TRAIL were not affected by AKI (data not shown). Please include the data in the manuscript.

Minor issues:

- The manuscript contains multiple grammatical errors that should be improved. Please have the manuscript proofread by a native English speaker.
- The manuscript, figure legends, and captions within the figures contain many abbreviations that are not spelled out fully the first time they are used or written out fully in the figure legends.
- The method section lacks essential details to reproduce the experiments successfully, for example:
 - o Native page electrophoresis, Co-IP, ChIP, cell viability assay: details on chemicals (e.g. article numbers/clones of antibodies, the composition of buffers such as the loading buffer, protease inhibitors, PMSF), concentrations (e.g. glycine, PMSF, protease inhibitor) and practical aspects, such as timing (e.g. chromatin sonication) is missing.
 - o Statistical analysis: a One-Way ANOVA was performed in case more than two groups were compared. Which post hoc test was performed to identify between-group differences?
- Figure 1K and L: it is unclear how these figures relate to each other. Figure 1K demonstrates cell death as measured by Annexin-V levels using flow cytometry, but the method used to obtain the numbers in Figure 1L is not clear to me. The percentages in figure 1K and 1L do not seem to match, which suggests that cell death in figure 1L is measured differently. This is, however, not clear from the figure or legend.
- Figure 1D and supplementary figure 1C: the sample size appears to be much higher in this analysis than, for example, the sample size used to measure gene expression (figure 1B). How many sections were measured per animal? Is the represented sample size actually the total number of sections analyzed, which could be multiple per animal? This approach would underestimate the biological variation if all these samples have been used to calculate the plotted variation.
- Figure 3A and 3B: the total lysate does not demonstrate different levels between control (Vec) and

DR5 overexpressed cells. What is the explanation for this? Similarly, the protein levels of DR5 and KIM1 are not higher after Cisplatin treatment than the control, which seems to contradict the results in figure 1.

- The caption in the figures and description in its legends is insufficient to understand the co-IP experiments' results easily. Which antibody was used for IP and for IB? In my view, the part of the blot that shows "IP: KIM1" (figure 3A/B) is confusing, as not all samples are IP'd using anti-KIM1 (half of the samples IgG as control). Does this part of the blot represent total protein at 70 kDa, or was KIM-1 immunoblotted? The same issue occurs in other figures presenting co-IP data analog to this example.
- Line 318: "exogenous overexpressed DR5". I doubt whether this is correct, as overexpression results in an increased expression of endogenous DR5 (i.e. derived or originated internally, from within the cells).
- Figure 7: injection of P2 partly prevents cell death and inflammation – as evidenced by gene expression of IL6 – in Cisplatin treated mice. In which matrix (e.g. saline, DMSO, ethanol?), the peptides were dissolved and what volume was injected is unclear. The dosage of P2 injected i.v. into the mice is unclear. Why was P2 injected i.p. and i.v.? Further, a control group treated with vehicle (e.g., saline, DMSO, or ethanol) group of mice are lacking.

Point-by-point Response Letter

The authors thank the reviewers for giving us the opportunity to submit a revised manuscript. We appreciate the insightful and constructive comments provided by the reviewers, which were helpful for improving the quality of our work. As requested, we have performed new experiments, re-analyzed some data and re-written some parts of the manuscript to address these comments. The revisions are shown in blue fonts for convenience. The following are our point-by-point responses.

REVIEWER COMMENTS

Reviewer #1 (Remarks to the Author):

This is a very interesting study which studied the YY1-KIM1-DR5 axis in the progression of AKI. The most important finding of this study is that YY1 transcriptionally regulates KIM1 and KIM1 interacts with DR5. This axis was demonstrated to be involved in the progression of AKI.

In this study has been clearly elaborated in this study, including those downstream pathways and phenotypic alterations. However, the mechanism of YY1 upregulation was less-well studied in the current study. As in the current models, both irradiation and cisplatin induced DNA damage, could DNA damage be a driving factor for YY1 upregulation.

Response: The authors appreciate the reviewer for the encouraging comment that “This is a very interesting study which studied the YY1-KIM1-DR5 axis in the progression of AKI”. To address the comment that “could DNA damage be a driving factor for YY1 upregulation”, we treated HK-2 cells with etoposide, a classic inducer of DNA damage, for different times (0, 5, 10, 15, and 30 minutes), and examined the mRNA levels of *YY1* and the DNA damage marker *P21*. The results showed that *P21* was upregulated at 5 minutes, while *YY1* was decreased at 10 minutes, after etoposide treatment, which was consistent with a recent report that YY1 is decreased upon DNA damage (Diabetes 2022 Aug 1;71(8):1694-1705). This new piece of data is provided in Supplementary Fig. 2e of the revised manuscript.

2. Since the YY1 inhibitor is available, it is very interesting to see if the YY1 inhibitor could reverse or alleviate the progression of AKI

Response: The authors thank the reviewer for the helpful suggestion. Since YY1 is decreased upon AKI, and, to our knowledge, YY1 inhibitors are not commercially available so far, we performed new experiments utilizing Eudesmin, a compound that was previously reported to upregulate YY1 expression (Sci Transl Med. 2019 Sep 18;11(510):eaaw2050), and tested its effects on cisplatin-induced AKI. YY1 was upregulated, while KIM1 was downregulated, in Eudesmin treated cisplatin-injured mice. Moreover, there was a significant reduction of serum creatinine and urea nitrogen levels as

well as a reduction in pathological injury in these mice. The new data are presented in Fig. 3f-j of the revised manuscript.

Consistently, overexpression of YY1 or Eudesmin treatment in HK-2 cells relieved cisplatin-induced cell death, and decreased the expression of inflammatory factors, while knockdown of YY1 aggravated cisplatin-induced cell death and increased the expression of inflammatory factors as we previously showed. Taken together, these results suggest that activating YY1 protects against AKI *in vitro* and *in vivo*.

3. YY1 has been extensively studied in many diseases, could the authors use the CHIP-seq data from other studies to study if the KIM1 was a potential target of YY1

Response: As the reviewer suggested, we searched the ChIP-Seq dataset from GTRD (Gene Transcription Regulation Database, GTRD) (<http://gtrd.biouml.org/>), which suggested that YY1 is enriched at multiple sites in the P3 and P4 regions of the *KIM1* promoter in HEK 293T and lymphoblastoid cells. This ChIP-Seq data agrees with our ChIP-qPCR results, and has been provided in Supplementary Table 1 of the revised manuscript.

4. The methodology is very sound, and enough details were provided, and I have no further suggestions for this part.

Response: The authors thank the reviewer for this encouraging comment.

Reviewer #2 (Remarks to the Author):

The manuscript investigated the regulation of kim-1 in the progression of acute kidney injury.

Comments

1) KIM-1 has been well investigated and validated as a marker of AKI. Both models of AKI, cisplatin and ischemia-reperfusion AKI were applied. Could authors explain any difference of the two models in the expression pattern/kinetic of KIM-1?

Response: As the reviewer suggested, we evaluated the transcriptional level of *Kim1* in two AKI models. We collected renal samples at Day 1, 3, 7 and 21 after unilateral ischemia-reperfusion injury as previously reported (J Clin Invest. 2019 Nov 1;129(11):4797-4816; Life Sci. 2020 Sep 1;256:117860). However, we only collected renal samples at Day 1 and Day 3 after cisplatin-injury, due to the high mortality (about 75%) at Day 4 in our experimental groups, which is consistent with other studies showing similar high mortality after Day 3 (Int J Mol Sci. 2019 Jun 20;20(12):3011; Am J Physiol Renal Physiol 313:F835–F841, 2017). Our results showed a similar expression pattern of *Kim1* in the two mouse AKI models, in which *Kim1* was dramatically upregulated at Day 1 after injury, decreasing afterward (Supplementary Fig. 2b of the revised manuscript).

However, the expression level of *Kim1* was significantly higher in the ischemia-reperfusion injury model at either Day 1 or Day 3 compared with cisplatin injury model, which may be the result of more severe renal damage in ischemia-reperfusion injury (Am J Pathol. 2012

Sep;181(3):818-28; Biochemistry (Mosc). 2019 Dec;84(12):1502-1512.), and also because the ischemia-reperfusion injury model involves only kidney specific injury.

2) It has been reported that KIM-1 levels increased after one-day post-cisplatin administration and this elevation correlated with AKI (Int. J. Mol. Sci. 2019, 20(20), 5238); Moreover, KIM-1 expression is at peak 24 h after IRI (doi: 10.7150/thno.73426). In the present study, for cisplatin model was intraperitoneally injected into 8-week-old male mice and euthanized on day 3 after the injections; for IRI, mice were euthanized at indicated day after injury to harvest blood and kidneys, but the time points were not well mentioned. The time points should be well selected to investigate the expression of KIM-1;

Response: The authors thank the reviewer for this kind reminder. Considering that KIM1 expression is highest at Day 1 after injury, and bilateral ischemia-reperfusion resulted in high mortality at Day 3 (Life Sci. 2020 Sep 1;256:117860), we performed bilateral ischemia-reperfusion injury on WT and *Kim1^{Ksp}* KO mice and collected samples at Day 1 after the injury. As suggested, the time points for bilateral ischemia-reperfusion injury are provided in the revised Method section and corresponding figure legends. For the cisplatin injury model, most samples were collected at Day 3 after the injury because the most severe pathological changes were found at this time point (Int J Mol Sci. 2019 Jun 20;20(12):3011; Kidney Int. 2022 May;101(5):987-1002.).

3) Transcription factor Yin Yang 1 (YY1), downregulated upon AKI, bound to the promoter of KIM1 and repressed its expression. Again, the expression pattern of YY1 should be further investigated. Is YY1 expression ahead of KIM-1?

Response: As the reviewer suggested, the transcriptional level of *Yy1* was examined along with *Kim1* level in both cisplatin- and ischemia-reperfusion-induced AKI mice models. The results showed that *Yy1* was dramatically decreased at Day 1 after injury when a significant increase in *Kim1* was found. The transcriptional level of *Yy1* started to be restored at Day 3 after injury when less *Kim1* was found, indicating transcriptional levels of *Yy1* and *Kim1* are negatively correlated in these two AKI mouse models (revised Supplementary Fig. 2b-d).

To investigate whether the change of YY1 takes place ahead of KIM1, we used cisplatin-treated HK-2 cells so that more time points could be obtained. The transcriptional level of *YY1* was significantly reduced at 3 hours and remained at a low level until 12 hours, and started to be restored at 24 hours after cisplatin treatment. The level of *KIM1* was upregulated at 6 hours and peaked at 12 hours, and then began to decrease at 24 hours after cisplatin treatment (Figure 2h of the revised manuscript). This new data indicates that the change of *YY1* level occurs ahead of *KIM1* upon injury.

4) In present study, authors draw the conclusion that KIM1 aggravates inflammation and apoptosis in renal tubular epithelial cells. A previous study (J Clin Invest. 2015; 125(4):

1620-36) reported KIM-1 protects the kidney after acute injury by downregulating innate immunity and inflammation. Only in vitro cell experiments were done in the present study to confirm this and further animal experiments should be done;

Response: As the reviewer suggested, we performed a new set of *in vivo* studies to investigate whether knockout of *Kim1* in renal tubular cells affects inflammation during injury. Compared to injured WT mice, renal tubular *Kim1* knockout reduced the serum IL-6 level after cisplatin- or ischemia/reperfusion- injury and inhibited macrophage infiltration (demonstrated by F4/80 staining) in these models (revised supplementary Fig 6), indicating that tubular KIM1 aggravates inflammation *in vivo*.

5) The expression of KIM-1 in Fig 1C, D was not consistent;

Response: The authors thank the reviewer for pointing out this issue. In the previous manuscript, we showed a long-time exposure (LE) picture of KIM1 to visualize the targeted band in all lanes. While a shorter exposure (SE) time showed a much lower KIM1 protein level in the uninjured group, which is consistent with KIM1 staining in Fig 1D. In the revised manuscript, both short and long exposure of KIM1 are provided in revised Fig 1C.

6) The experimental groups should be well labeled and noted in figure legends, e.g., CT, CT+Ctrl, Kim1^{ksp} KO (transgenic model only or with AKI, time points), et al.

Response: We have carefully checked and relabeled the relevant figures and figure legends throughout the revised manuscript as suggested.

Reviewer #3

The paper “A renal YY1-KIM1-DR5 axis regulates the progression of acute kidney injury” by Chen Yang et al. investigates the role of the interaction between DR5 and KIM1 in acute kidney injury. Specific rational designed peptides able to inhibit the interaction between DR5 and KIM1 were also able to reduce the renal toxicity and cytotoxicity induced by cisplatin treatment.

The topic is certainly important and deserves the attention of translational research. However, this work has many limitations. Although many different methodologies have been employed, in some parts the results are unclear.

Meanwhile, it is not conclusive about the message. That is, is the interaction between DR5 and KIM1 causal or not with respect to the development of renal failure?

Response: The authors thank the reviewer for this insightful comment. In our work, interaction between KIM1-DR5 was enhanced upon cisplatin stimulation in HK-2 cells, with enhanced co-localization of KIM1 with DR5 upon AKI injury. Rationally designed peptide P2 which disrupts the interaction between KIM1-DR5, relieved cisplatin-induced renal or tubular cell injury *in vivo* and *in vitro*. All these data clearly indicate that the KIM1-DR5 interaction plays an important role in the progression of AKI. However, to fully

address whether the interaction between DR5-KIM1 is causal in the development of AKI, study on knock-in mice that express KIM1/DR5 construct(s) with compromised interface between DR5 and KIM1 will be an interesting future direction as we discussed in the revised manuscript.

In my opinion, the work is missing a lot of important information that needs to be added:

1. Which alterations induces per se the overexpression of KIM1 in HK-2 cells such as to sensitize them to the treatment with cisplatin? For example, at the level of cytokine production, NF- κ B (Nuclear Factor kappa B) activation, DR5 expression and/or dimerization, mitochondrial function, oxidative stress?

Response: To address this comment, we treated HK-2 cells with IL-6 (mimicking cytokine production; 50 ng/mL), TNF- α (mimicking NF- κ B activation, 20 ng/mL), and H₂O₂ (mimicking mitochondrial dysfunction and oxidative stress, 800 μ M), alone or in combination with 5 μ g/mL cisplatin. The mRNA level of *KIM1* was up-regulated in response to cisplatin, or TNF- α or IL-6 or H₂O₂ *per se*. Moreover, IL-6 exhibited a synergetic effect with cisplatin on *KIM1* up-regulation, indicating that cytokine production may also contribute to the upregulation of KIM1 upon renal injury. This piece of new data is provided in revised Supplementary Fig. 1f.

2. Is there a change in the molecular interaction between KIM1 and DR5 in KIM1 overexpressing cells?

Response: The authors thank the reviewer for this insightful comment. To detect the interaction between DR5 and gradually overexpressed KIM1 by FRET, we co-transfected HK-2 cells with DR5-YFP (1 μ g) and an increased amount of KIM1-CFP (0, 0.2, 0.4, 0.6, 0.8, and 1 μ g). To further detect the multimerization of DR5 by FRET, we co-transfected HK-2 cells with DR5-CFP (1 μ g), DR5-YFP (1 μ g) and an increased amount of KIM1 (0, 0.2, 0.4, 0.6, 0.8, 1 μ g). KIM1-CFP promoted the interaction between KIM1-DR5 in a concentration dependent manner, as indicated by a gradually increased FRET intensity with increasing amount of KIM1-CFP. Moreover, KIM1 overexpression also promoted DR5 multimerization in a concentration dependent manner, as indicated by a gradually increased FRET intensity between DR5-CFP and DR5-YFP with increasing amount of KIM1. Together, our data indicated that KIM1 overexpression promotes its interaction with DR5 and consequently promotes the multimerization of DR5. These pieces of new data are presented in Supplementary Fig. 3b & 3g of the revised manuscript.

3. In Materials and Methods section, the authors describe only the conditions related to FRET experiments to study the interaction between exogenous KIM1 (KIM1-HA) with overexpressed Flag-tagged DR5 in HK-2 cells. I have not found information on how the FRET experiments were carried out to study the interaction between endogenous KIM1

and DR5, which in my opinion are much more important. Were specific antibodies used. by FRET intensity do the authors mean the FRET efficiency?

Response: The authors thank the reviewer for this comment. FRET assay is a classic method to probe the interaction between proteins (Methods Cell Biol. 2008;85:381-93.; Methods Mol Biol. 2015;1251:67-82), which relies on fluorescent labeling of molecules without using an antibody. In our experiments, the interaction between CFP-tagged KIM1 and YFP-tagged DR5 was detected by a standard FRET assay, in which the FRET signal is detected when two fluorescent molecules are in close proximity (< 10 nm). We have rewritten the relevant Method and Results sections to describe how the FRET assays were performed and analyzed.

Quantitative analysis of FRET intensity was achieved by fluorimeter-based detection, which recorded the FRET signal by the excitation of CFP and emission of YFP. A strong FRET intensity indicates strong interaction between detected molecules. FRET efficiency was used to quantitate acceptor photobleaching based FRET assays. Photobleaching of DR5-YFP (acceptor) leads to increased fluorescence intensity of DR5-CFP (donor). FRET efficiency was calculated by the fluorescence intensity of the donor (DR5-CFP) before (I_{pre}) and after (I_{post}) the photobleaching with the formula: $E_{FRET} = (1 - I_{pre}/I_{post})$ (Biophys J. 2004 Jun;86(6):3923-39; Front Mol Biosci. 2021 May 14;8:635548). A high FRET efficiency indicates a strong interaction between the detection molecules.

4. Nystatin is not a DR5 dimerization inhibitor rather a lipid raft disorganizer. Since, as reported in previous papers, DR5 transmits the apoptotic signal only when localized in the rafts, the effect observed by the authors following pre-treatment with nystatin seems to depend on this aspect. Authors should try to use methyl- β -cyclodextrins or statins, which have the advantage of being less toxic than nystatin both *in vitro* and *in vivo* and perifosine, which, on the contrary, induced the recruitment of DR5 into lipid microdomains.

Response: As the reviewer suggested, we performed new FRET assays to evaluate whether compounds that affect lipid rafts, such as atorvastatin (inhibiting lipid accumulation (Int Immunopharmacol. 2010 Aug;10(8):892-9) and perifosine (recruiting DR5 into lipid microdomains (Blood 2007 Jan 15;109(2):711-9; Cell Death Dis. 2013 Oct 17;4(10):e863)), affect DR5 multimerization *in vitro*. However, there was no obvious effect on DR5 multimerization after either atorvastatin or perifosine treatment. The new data are presented in Supplementary Fig. 4f-g. These data suggest that Nystatin may affect DR5 dimerization not only due to its function as a lipid raft disorganizer.

5. Many papers report an increase in apoptosis in AKI models both *in vitro* and *in vivo*. All that, even though different ways, can reduce apoptosis has been found to be effective in improving renal pathological injury. Among these: antioxidants and anti-inflammatories of various kinds, mitochondrial protectors, inhibitors of the opening of the mitochondrial pore, and pure inhibitors of apoptosis such as over-expression of Bcl-2. This further highlight

that apoptosis represents a key event in the onset and evolution of the disease, at least that linked to drug toxicity, but equally underlines how a *primum movens* that could indicate a promising and innovative therapeutic target has not yet been identified.

Response: The authors thank the reviewer for the comments. We addressed this issue in the Discussion section (page17) of the revised manuscript.

Reviewer #4 (Remarks to the Author):

The authors describe the downregulation of YY1 upon Cisplatin or I/R-induced kidney injury to lead to an increase of KIM-1 in the kidney. In turn, upregulation of KIM-1 binds to DR-5, leading to gene expression of pro-inflammatory cytokines and induction of cell death. The authors convincingly demonstrate the effect of overexpression and knock-out of KIM-1 on relevant targets in the YY-1-KIM1-DR-5 pathway, as well as downstream inflammation and cell death. Experiments in Cisplatin-treated mice and cells and the induction of I/R in mice support the conclusion. The concise set of experiments seems to be well-designed and follows a clear strategy, using state-of-the-art methodology. The results sufficiently support the work's findings. The results are novel and noteworthy:

KIM1 is a well-known biomarker for AKI and current results now demonstrate a mechanistic role of KIM1 in the pathogenesis of AKI, which are noteworthy results.

Response: The authors thank the reviewer for the comments that “The concise set of experiments seems to be well-designed and follows a clear strategy, using state-of-the-art

methodology. The results sufficiently support the work's findings. The results are novel and noteworthy".

However, I have identified several inconsistencies and textual issues that should be corrected prior to publication. Amongst others, additional data is required to understand the interaction sites between KIM1 and DR5, relate the protective effects of KIM1Ksp KO mice to the extent of AKI in WT mice with and without Cisplatin, and explain the protective effects of the P2 peptide. The method section lacks some details that preclude the reproduction of some of the experiments.

Response: The authors thank the reviewer for the insightful suggestions. We performed new domain mapping experiments to identify the binding sites between KIM1 and DR5 as the reviewer suggested. As suggested, we also provided more detailed information in the revised Methods section, and corrected the main text and legends accordingly.

Major issues:

The authors used different human kidney cell lines, namely HK-2, HEK-293, and 293T.

What is the underlying reason, and to what extent may this influence the results?

Response: The authors thank the reviewer for this reminder. In this study, two human renal cell lines were used. HK-2 is a widely used immortalized proximal tubule epithelial cell line. HEK293T (293T), a human embryonic kidney cell line, is a widely used tool cell line

to study protein-protein interaction, *etc* (Nat Commun. 2017 Dec 18;8(1):2164; Cell. 2021 May 27;184(11):3022-3040.e28). In our study, KIM1 was either overexpressed or knocked out in HK-2 cells to study its function in the regulation of inflammation, apoptosis and DR5 multimerization *in vitro*. In HEK293T cells, proteomics and luciferase reporter assays were performed.

The authors developed an antagonistic peptide (P2) that blocked KIM1-DR5 interaction and exhibited reno-protective effects *in vivo* and *in vitro*. To interact with KIM1-DR5, the peptide needs to be targeted to the intracellular milieu of tubular cells. It is unclear from the manuscript how the authors envision the peptide ending in the cells expressing KIM1-DR5. Can the authors demonstrate the localization of the P2 peptide into the target cells?

Response: The authors thank the reviewer for this insightful suggestion. To investigate whether peptide P2 targets to cells overexpressing KIM1-DR5, we performed the following *in vitro* and *in vivo* studies.

In Vitro study: We labeled peptide P2 with FITC, and incubated labeled P2 with cisplatin treated HK-2 cells, then immunofluorescent stained for KIM1 and DR5. Increased KIM1 and DR5 staining were found after cisplatin injury with FITC-labeled P2 bound to injured cells, indicating that peptide P2 targets to cells highly expressing KIM1-DR5 (revised Supplementary Fig. 8a).

In vivo study: We labeled peptide P2 with Cy7, a near-infrared fluorescein, and performed *ex vivo* organ imaging in cisplatin-injured mice. The results indicated that Cy7 labeled P2 (1 μ M) exhibited a dramatically increased renal distribution compared with the control (free Cy7 only), suggesting that peptide P2 accumulates in the injured kidney (revised Supplementary Fig. 8b). Moreover, we labeled peptide P2 with 5'(6)-FAM, which is a green fluorescein that exhibits better *in vivo* stability compared with FITC. 5'(6)-FAM-labeled P2 was intravenously injected into cisplatin-injured mice. KIM1 and DR5 were immunofluorescent stained on renal cryo-sections and merged with the image of 5'(6)-FAM-labeled P2. These results demonstrated that labeled P2 co-localized with KIM1 and DR5 in the renal tubules of cisplatin-injured mice (Revised Fig. 8e).

Together, all of the new data suggest that peptide P2 is targeted to cells overexpressing KIM1-DR5 *in vivo* and *in vitro*.

The rationale for the dosage of P2 in mice is lacking. How did the authors decide on this dose? Was this extrapolated from the *in vitro* experiments? If so, how? P2 only partially prevented Cisplatin-induced cell death in the kidney, with only marginal effects on serum creatinine and BUN. Could this observation be due to relatively low tissue levels of P2? The manuscript could be improved by a) demonstrating levels of P2 in the kidney and b) demonstrating the effect of a higher dosage of P2 on the kidney upon Cisplatin treatment.

Response: The authors thank the reviewer for pointing out this issue. Indeed, we performed a pilot dosage study of peptide P2 (0.2, 1, 2 and 5 μM) in cisplatin-injured mice. Renal function assessments and H&E staining results suggested that all tested dosages (0.2, 1, 2 and 5 μM) of peptide P2 exhibited reno-protective effects against cisplatin injury (revised Supplementary Fig. 7d-f). Of these, 1 μM exhibited better reno-protective effects compared with 0.2 μM of peptide P2 as indicated by urea nitrogen levels (revised Supplementary Fig. 7e). Since our pilot results did not suggest that higher dosages of peptide P2 (2 or 5 μM) exhibited better therapeutic effects 1 μM of peptide P2 was used for *in vivo* studies.

- The results from *in silico* mapping of the Interaction sites (result: ECD deletion abolished interaction) and *in vitro* co-IP experiments (result: KIM1 binds to TMH) seem to contradict each other. How can the authors explain this?

Response: Our results from *in silico* mapping are consistent with *in vitro* Co-IP. ECD deletion abolished the KIM1-DR5 interaction as suggested by molecular dynamic simulations (Supplementary Table 3); while in the Co-IP assays, DR5 Δ ECD failed to bind with KIM1 (Fig 51). Both results indicate that KIM1 binds to the ECD domain of DR5.

- The authors demonstrate that KIM1 binds to the TMF-domain of DR5. However, it remains unclear which domain of KIM1 interacts with the TMH region in DR5. Can the authors demonstrate the binding domain of KIM1? The authors present a drawing on the

interaction between KIM1-DR5 (Figure 7L) that does not seem to represent the findings in the manuscript. Although the authors demonstrate that KIM1 seems to bind TMH on DR5 (or ECG, based on in silico results?) they do not seem to provide evidence to support the binding domain of KIM1. Instead, the figure suggests that the cytosolic domain of KIM1 binds to the CytD domain of DR5, which partly contradicts their results and is also not fully supported by data. Although I am not sure whether the authors intended to demonstrate such a detailed view of the interactions between both proteins, I believe the manuscript could best be enriched with data revealing the interacting domains of KIM1 and DR5.

Response: The authors thank the reviewer for this great suggestion. Accordingly, we constructed a set of plasmids expressing individual KIM1 regions, which includes Flag-tagged KIM1 Ig V, KIM1 Mucin, KIM1 TM + CytD, and performed new Co-IP assays. The results indicated that DR5 binds to WT KIM1 and KIM1 Ig V, suggesting that the Ig V domain is crucial for the binding with DR5. The new data is presented in revised Fig. 5m. Furthermore, the interaction between KIM1 and DR5 was re-drawn in the revised Fig. 8l as the reviewer suggested.

- The authors demonstrate lowered levels of serum creatinine, BUN, reduced gene expression of pro-inflammatory genes, and lower levels of cell death in KIM1^{Ksp} KO animals treated with Cisplatin as compared to WT animals treated with Cisplatin (Figures

5 and 6). Yet, the authors do not demonstrate such values in WT animals not treated with Cisplatin, which makes it impossible to interpret the relevance of the findings beyond a proof-of-concept. To what extent does KIM1Ksp KO protect against AKI in Cisplatin-treated mice? The levels are lower than WT mice treated with Cisplatin, but how does this relate to healthy mice? The same issue applies to the lack of a control group for the I/R experiments in mice (Figure 6).

Response: The authors thank the reviewer for the suggestions. As suggested, we re-organized the data. In the revised Fig. 6, the levels of serum creatinine and BUN, expression of pro-inflammatory, apoptotic and fibrotic genes were shown together in WT and *Kim1*^{Ksp} KO mice with or without cisplatin injury.

- The authors note that TRAIL can bind and activate DR5 in the discussion, and mention that the levels of TRAIL were not affected by AKI (data not shown). Please include the data in the manuscript.

Response: As suggested, we included the mRNA and protein levels of TRAIL in cisplatin- and uIRI- injured mouse models in the revised Supplementary Fig. 3c-f.

Minor issues:

- The manuscript contains multiple grammatical errors that should be improved. Please have the manuscript proofread by a native English speaker.

Response: The authors thank the reviewer for the suggestion. The revised manuscript has been edited by Prof. Robert Petersen, a coauthor and a native English speaker.

- The manuscript, figure legends, and captions within the figures contain many abbreviations that are not spelled out fully the first time they are used or written out fully in the figure legends.

Response: We have re-checked the manuscript to make sure that the abbreviations are spelled out fully at the first time they were used.

- The method section lacks essential details to reproduce the experiments successfully, for example: Native page electrophoresis, Co-IP, ChIP, cell viability assay: details on chemicals (e.g. article numbers/clones of antibodies, the composition of buffers such as the loading buffer, protease inhibitors, PMSF), concentrations (e.g. glycine, PMSF, protease inhibitor) and practical aspects, such as timing (e.g. chromatin sonication) is missing.

Response: We have re-checked the Method section and did our best to include experimental details as the reviewer suggested.

Statistical analysis: a One-Way ANOVA was performed in case more than two groups were compared. Which post hoc test was performed to identify between-group differences?

Response: The authors thank the reviewer for the helpful comment. Tukey's multiple-comparison test was performed as a post hoc test to identify between-group differences in our study. We have provided this information in the revised Method section and relevant figure legends.

- Figure 1K and L: it is unclear how these figures relate to each other. Figure 1K demonstrates cell death as measured by Annexin-V levels using flow cytometry, but the method used to obtain the numbers in Figure 1L is not clear to me. The percentages in figure 1K and 1L do not seem to match, which suggests that cell death in figure 1L is measured differently. This is, however, not clear from the figure or legend.

Response: Sorry for the confusion caused by the data presentation. To present this data more clearly, we re-organized the figure panels so that the quantitative data is now shown in the same figure panel together with the flow results. The reason that the percentages in the previous figures 1K and 1L did not seem to match was because that figure for flow was a representative result while the quantitative data provided the average and standard deviation for three independent experiments (percentage of apoptotic cells was calculated by Annexin-V positive cells (Q2+Q3)). In the revised manuscript, this information has been clearly described in the revised Method section and the corresponding figure legends.

- Figure 1D and supplementary figure 1C: the sample size appears to be much higher in

this analysis than, for example, the sample size used to measure gene expression (figure 1B). How many sections were measured per animal? Is the represented sample size actually the total number of sections analyzed, which could be multiple per animal? This approach would underestimate the biological variation if all these samples have been used to calculate the plotted variation

Response: The authors thank the reviewer for this kind reminder. In Fig. 1d and the supplementary Fig. 1c of the previous manuscript, we quantified the IHC staining by fields rather than by n-numbers. In the revised manuscript, we have re-quantified the IHC staining results by n-numbers and reached the same conclusion (revised Fig. 1d, Fig. 4f and supplementary Fig. 1c).

- Figure 3A and 3B: the total lysate does not demonstrate different levels between control (Vec) and DR5 overexpressed cells. What is the explanation for this? Similarly, the protein levels of DR5 and KIM1 are not higher after Cisplatin treatment than the control, which seems to contradict the results in figure 1.

Response: The authors thank the reviewer for the insightful comment. We have re-performed the Co-IP assays with more convincing results shown in the revised Fig. 4a-b.

- The caption in the figures and description in its legends is insufficient to understand the co-IP experiments' results easily. Which antibody was used for IP and for IB? In my view,

the part of the blot that shows "IP: KIM1" (figure 3A/B) is confusing, as not all samples are IP'd using anti-KIM1 (half of the samples IgG as control). Does this part of the blot represent total protein at 70 kDa, or was KIM-1 immunoblotted? The same issue occurs in other figures presenting co-IP data analog to this example.

Response: The authors thank the reviewer for this helpful suggestion. We have re-checked and revised all figure panels and corresponding legends related to Co-IP assays as suggested.

- Line 318: "exogenous overexpressed DR5". I doubt whether this is correct, as overexpression results in an increased expression of endogenous DR5 (i.e. derived or originated internally, from within the cells).

Response: The authors thank the reviewer for the comment, we have revised the sentence to read as “A co-IP study demonstrated interaction between KIM1 and DR5, with or without DR5 overexpression, at both physiological conditions and under cisplatin injury”.

- Figure 7: injection of P2 partly prevents cell death and inflammation – as evidenced by gene expression of IL6 in Cisplatin treated mice. In which matrix (e.g. saline, DMSO, ethanol?), the peptides were dissolved and what volume was injected is unclear. The dosage of P2 injected i.v. into the mice is unclear. Why was P2 injected i.p. and i.v. Further, a

control group treated with vehicle (e.g., saline, DMSO, or ethanol) group of mice are lacking.

Response: The authors thank the reviewer for the kind reminder. In our study, peptide P2 was dissolved in 10 mM PBS containing 1% DMSO, and was i.v. injected. P2 peptide was only injected by i.v. in this study, while the control mice (CT) were i.v. injected with the same volume of 10 mM PBS containing 1% DMSO. We have included the information in the revised Method section (Pages 19-20).

REVIEWERS' COMMENTS

Reviewer #1 (Remarks to the Author):

The authors have addressed the concerns of reviewers, I have no further comments.

Reviewer #2 (Remarks to the Author):

The authors addressed the reviewer's comments well.

Reviewer #3 (Remarks to the Author):

Dear Authors,

after revisions, the paper is significantly improved and the issues I raised have all been addressed.

However, one of the questions raised under point 3 has not been answered.

As regards point 3, I wanted to understand if the authors had investigated the molecular association between KIM and DR5 ion cells not overexpressing the proteins. Studying protein interactions between endogenous proteins rather than in overexpressing cells I believe is more correct and informative. In the case of endogenous proteins, FRERT could be done using specific antibodies and, such as fluorochromes.

Reviewer #4 (Remarks to the Author):

The authors used the feedback provided by me and the other reviewers to improve the content and readability of the manuscript. Every comment made by me in response to the previous version, has been appropriately addressed by the authors. I have no further comments on this manuscript.

Point-by-point Response Letter

We sincerely thank the referees for their suggestions. We are glad to know that Reviewers 1, 2, and 4 thought our revisions have adequately addressed their comments. In this letter, we addressed the remaining question by Reviewer 3. The changes are shown in blue fonts in the revised manuscript for convenience. Following are our point-by-point responses.

Reviewer 3

After revisions, the paper is significantly improved and the issues I raised have all been addressed. However, one of the questions raised under point 3 has not been answered. As regards point 3, I wanted to understand if the authors had investigated the molecular association between KIM and DR5 in cells not overexpressing the proteins. Studying protein interactions between endogenous proteins rather than in overexpressing cells I believe is more correct and informative. In the case of endogenous proteins, FRERT could be done using specific antibodies and, such as fluorochromes.

Response: The authors thank the reviewer for the encouraging comment that “After revisions, the paper is significantly improved and the issues I raised have all been addressed”. We agree with the reviewer that studying endogenous KIM1-DR5 interaction is important. In this work, endogenous interaction of KIM1 and DR5 has been studied by Co-IPs in HK-2 cells (Fig. 4b), as well as by co-staining of KIM1 and DR5 in HK-2 cells and mouse renal sections (Fig. 4i-j). In our study, FRET assays are performed based on fluorescent pairs CFP and YFP, which are exogenous constructed to KIM1 and DR5.

Actually, FRET assay is not an ideal approach for endogenous protein interactions detection, which detects protein interactions within 10 nm (Nat Biotechnol. 2003 Nov;21(11):1387-95.; Chem Soc Rev. 2014 Sep 7;43(17):6370-404.). The strict distance requirement makes FRET difficult to probe endogenous KIM1-DR5 interaction by specific antibodies like fluorochromes, since the size of a single primary antibody *per se* is about 10 nm. Therefore, we have included this as a limitation in the revised manuscript as suggested. As future studies, endogenous FRET assays may be performed by engineering cells using CRISPR/Cas9 mediated knock-in system (Mol Cell. 2021 Mar 18;81(6):1216-1230.), so that KIM1 and DR5 are labeled endogenously with CFP or YFP, respectively (Page 10 of the revised manuscript).